# Spatiotemporal evaporating droplet dynamics on fomites enhances long term bacterial pathogenesis

Sreeparna Majee [1], Atish Roy Chowdhury[2], Roven Pinto[1,4], Ankur Chattopadhyay[1,4], Amey Nitin Agharkar[3], Dipshikha Chakravortty [2✉] & Saptarshi Basu [1,3✉]

Naturally drying bacterial droplets on inanimate surfaces representing fomites are the most consequential mode for transmitting infection through oro-fecal route. We provide a multi-scale holistic approach to understand flow dynamics induced bacterial pattern formation on fomites leading to pathogenesis. The most virulent gut pathogen, *Salmonella* Typhimurium (STM), typically found in contaminated food and water, is used as model system in the current study. Evaporation-induced flow in sessile droplets facilitates the transport of STM, forming spatio-temporally varying bacterial deposition patterns based on droplet medium's nutrient scale. Mechanical and low moisture stress in the drying process reduced bacterial viability but interestingly induced hyper-proliferation of STM in macrophages, thereby augmenting virulence in fomites. In vivo studies of fomites in mice confirm that STM maintains enhanced virulence. This work demonstrates that stressed bacterial deposit morphologies formed over small timescale (minutes) on organic and inorganic surfaces, plays a significant role in enhancing fomite's pathogenesis over hours and days.

[1] Department of Mechanical Engineering, Indian Institute of Science, Bangalore 560012, India. [2] Department of Microbiology and Cell Biology, Indian Institute of Science, Bangalore 560012, India. [3] Interdisciplinary Centre for Energy Research (ICER), Indian Institute of Science, Bangalore 560012, India. [4]These authors contributed equally: Roven Pinto, Ankur Chattopadhyay. ✉email: dipa@iisc.ac.in; sbasu@iisc.ac.in

Droplets settled on surfaces of fomites are considered as one of the most significant modes of microbial infection transmission[1,2]. The surface of fomites contaminated with bacterial pathogens is found to be responsible for the outbreak of community-acquired and nosocomial infections. Fomites serve as a reservoir for pathogenic microorganisms, which transmit diseases caused by bacteria including healthcare facilities[3]. Moreover, animal studies have shown that gavage with faecal supernatant or exposure to faecal fomites can infect a large proportion of animals[4]. High concentrations of fecal indicator bacteria (FIB) and pathogens are found in food, water, soil, in addition to hands and commonly available surfaces[5–7]. Inadequate sanitation leads to augmented fecal contamination levels in the environment causing malnutrition, stunting, and even morbidity and mortality from diarrheal diseases[8]. Moreover, fecal contamination on surfaces is associated with an increased risk of diarrheal disease, as highlighted by a study at a child care center in the United States[9].

*Salmonella* Typhimurium is (STM) one of the most virulent bacterial pathogens that frequently cause food-borne gastroenteritis in humans and its infection continues to remain a serious health hazard in Asian and African countries[10]. Stanaway et al.[11] estimated that 535,000 (95% Uncertainty Interval 409,000–705,000) cases of non-typhoidal *Salmonella* invasive disease occurred in 2017, which caused 77,500 (46,400–123,000) deaths and 4.26 million (2.38–7.38) DALYs (disability-adjusted life-year), globally. Moreover, *S.* Typhimurium infects mice too, which has similar symptoms as typhoid fever caused by *S.* Typhi in humans[10]. Despite being a gut pathogen and having a fixed orofecal transmission route via contaminated food and water, many studies have demonstrated the aerosol transmission of specific serovars of *Salmonella* Enterica such as *Salmonella* Typhimurium, *Salmonella* Agona, etc. in poultry animals[12,13]. Aerosol mediated transmission of STM happens from contaminated feces and that can further contaminate poultry birds after 2–4 hrs of exposure[14]. Thus, settling of the bacteria on fodder and other surfaces leads to infection through ingestion causing food poisoning in poultry animals[10]. *Salmonella* Enteritidis, one of the leading causes of salmonellosis in poultry birds, can survive on the outer shell of contaminated eggs at a low temperature and low relative humidity in the presence or absence of any nutrients[15,16]. The improper sterilization of the medical equipment which is already contaminated with bacterial fomites is one of the major reasons behind healthcare-associated outbreaks of *Salmonella*[17]. The ability of STM to survive on inanimate surfaces such as polypropylene, formica, stainless steel, and wooden surfaces in the presence or absence of the protein source at room temperature, 6 hrs post-inoculation enhances the risk of contamination of consumable food items from the contact surface[18]. All these factors contribute to the significance of study of STM in droplets settled on fomites.

When an infected person sheds feces in public toilets, flushing generates aerosol, which may get deposited on bathroom surfaces in the form of droplets. Any healthy person coming in contact with such fomites can get infected via the oral transmission route. Moreover, the irrigation involves supply of water for producing the crops and vegetables. During water supply, droplets and aerosols of water can form, which can settle on different parts (such as leaf, stem, etc.) of a given plant. Assume that the water is already contaminated with *Salmonella*. Now this microbe can adhere to plant leaves, and even on end product (e.g., tomato[19]). This pathogen can transmit to humans and animals if the raw product is consumed without further processing (Supplementary Fig. 1). However, the cause of infection of such droplets deposited on surfaces was unclear. Therefore, understanding the dynamics of these bacteria-laden droplets found in the environment becomes necessary to unfold such fomite mediated pathogenesis. The fluid dynamics-based investigation contemplates bacterial droplets under evaporating conditions which finally form dried precipitates. The patterns formed due to agglomeration of bacteria on such dried precipitates highly depend on the nutrition scale of the fluid medium, comprising the droplets. The loss of fluid during evaporation is not an ideal environment for the bacteria to survive and generates stress in bacterial population[20]. However, STM can survive on inanimate dry surfaces from 10 days to 4.2 years[21]. The survival of the bacteria in the harsh environment of the dried fomite precipitate can be attributed to the presence of two components systems (TCS). The sensor kinase of TCS enables *Salmonella* to sense the environment, making them adapt and survive under stressed conditions by switching on the expression of appropriate response regulator genes[10]. Our study revealed that mechanical and low-moisture stress on the bacteria generated from evaporation dynamics in a droplet is the most significant factor that improved bacterial survival in murine macrophages. In both neutral and nutrient-rich media's dried precipitate, bacteria show comparable or enhanced infectivity than in liquid phase bacterial culture, even though viability is significantly reduced, undermining the effect of nutrition in infectivity. In the mouse model of infection, the comparable burden of planktonic bacteria with respect to the *Salmonella* retrieved from fomite precipitate suggested that even on dried surface, bacteria maintains virulence. Commonly found real surfaces such as steel door handle, tomato, and cucumber skin also confirms enhanced bacterial infectivity. This pioneering study demonstrates that the evaporation-flow coupled stress on bacteria, makes physical contact with dried fomites highly infectious and can cause further transmission of diseases. Figure 1 offers the overall schematic of the lifecycle of a fomite infection. The fluid dynamics dominant short time-scale, where bacterial transport, agglomeration and dehydration occurs, leads to the long time-scale of reduced viability but enhanced infection in any animal via oro-fecal route.

## Results

**Deposition patterns of bacteria in fomites**. To understand the pattern formation caused by bacterial agglomeration in an evaporating fomite under environmental factors, we have considered a sessile mode of droplet evaporation (experimental details are given in Methods section). To broaden our scope of experiments, different types of physiological fluid medium are taken into consideration to mimic real-life circumstances. Several media are considered viz. saline, dextrose, and mucin to emulate respiratory fluid, intravenous drip, and gastro-intestinal medium[22], respectively. STM wild-type (WT), STM ΔfliC, and PFA fixed dead bacteria are incorporated in different media to understand the flow patterns and their geometrical mechanics. Neutral media like saline and milli-Q are considered to apprehend nutrient based differences with nutrient-rich dextrose, and mucin media. We have used confocal microscope to examine the deposition patterns within the sessile droplet with the four different media (Fig. 2a) under 10× magnification. The bacteria, endogenously tagged with red fluorescent proteins (RFP), show contrasting deposition patterns after evaporation of the droplets ($t_f \approx 3-7$ mins), where $t_f$ is the total evaporation time. The inter-medium comparisons reveal dense and narrow bacterial edge depositions in saline and milli-Q, whereas, for dextrose and mucin, the edge is much more diffused. The edge deposits, similar to coffee ring[23], formed by capillary flow (Supplementary Movie 1), result from the bacterial aggregation due to enhanced evaporation flux at the three-phase contact line. In case of milli-Q and saline droplets, during evaporation, simultaneous pinning and de-pinning of

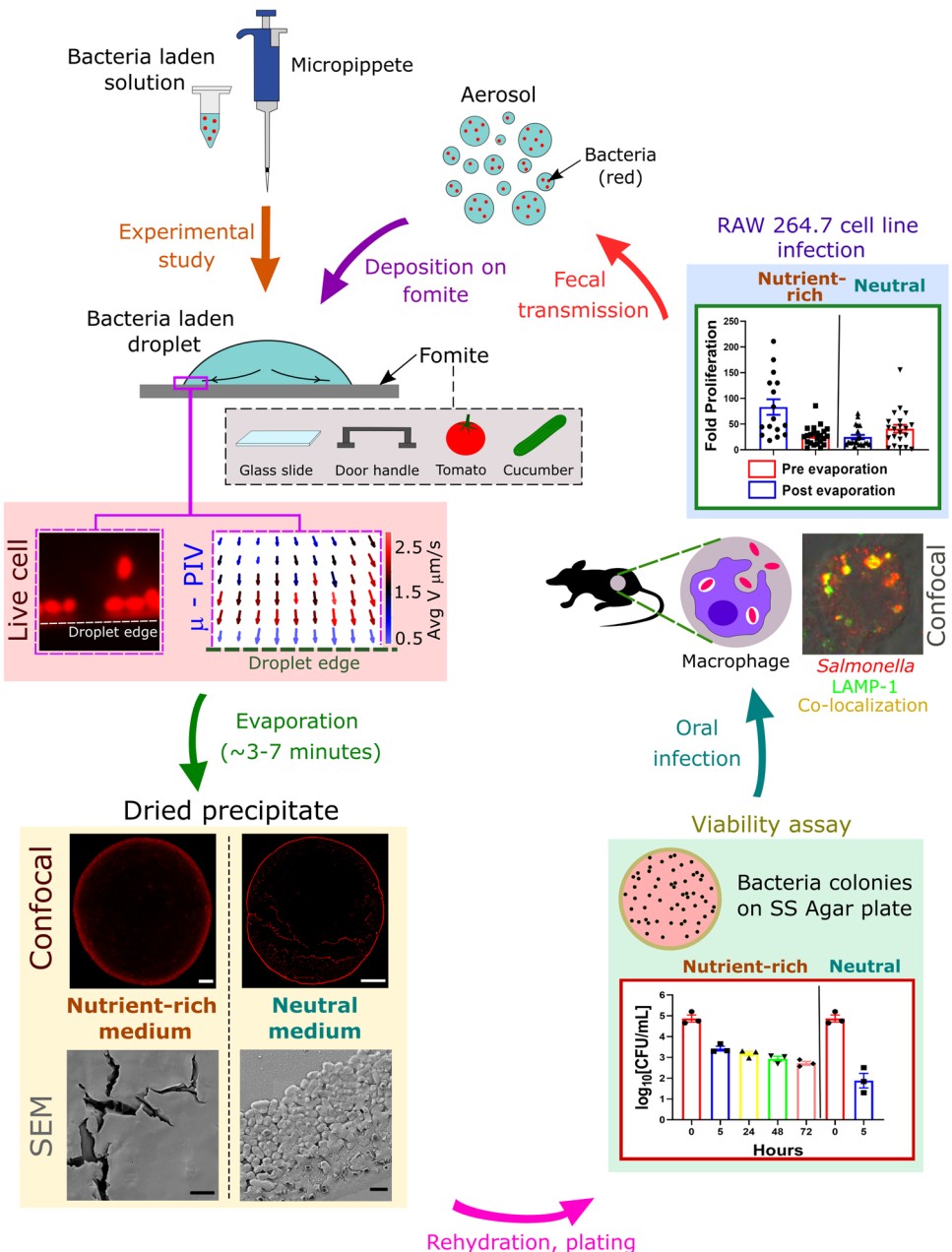

**Fig. 1 Schematic representation of the experimental study.** This involves live cell imaging, μ-PIV, confocal imaging, SEM, viability, and cell line infection. White scale bars, 100 μm and black scale bars, 2 μm.

contact lines are observed several times from the initial contact area as the final solvent depletes from a concentrated domain near the droplet periphery. The bacterial deposits are found at these intermediate pinned contact lines and at the final concentrated location, where the final stage of drying occurs. Based on the above observations, it is clear that neutral media viz. saline solution and milli-Q water have similar bacterial deposition patterns that differ significantly from the nutrient-rich media viz. dextrose and mucin. The quantitative analysis of the droplet precipitates for neutral solutions reveals denser edge deposition around the droplet periphery. However, in case of nutrient-rich media (dextrose and mucin), bacteria-laden droplets remained pinned for the entire duration of evaporation and the dried precipitates exhibit graded bacterial accumulation, unlike neutral media. Figure 2b, c depict the intensity profiles for saline and milli-Q solutions along the radial direction toward the center in

the peripheral region. The high-intensity regions confirm fluorescence signals from the bacterial deposition which shows denser edge deposition for STM WT and STM ΔfliC of ~7 μm (Fig. 2b, c) and discontinuous and wider deposition for dead bacteria of around 20 μm. On the contrary, for nutrient-rich media, the final deposition patterns at the edge are diffused over a larger area. To understand this difference, further investigations are carried out with dilute concentrations (dextrose 0.9 wt% and mucin 0.1 wt%) for both nutrient-rich media. Profilometry data (Fig. 2d, e) gives us a vivid idea about the deposition length scale, where the concentrated solutions have more extensive and broader edge deposition profiles than their low concentration counterparts; that prove that the edge deposition is influenced by solute concentration. Moreover, apart from glass slide, bacterial deposition pattern on realistic surfaces such as steel door handle and organic surfaces like tomato and cucumber skin are also studied with

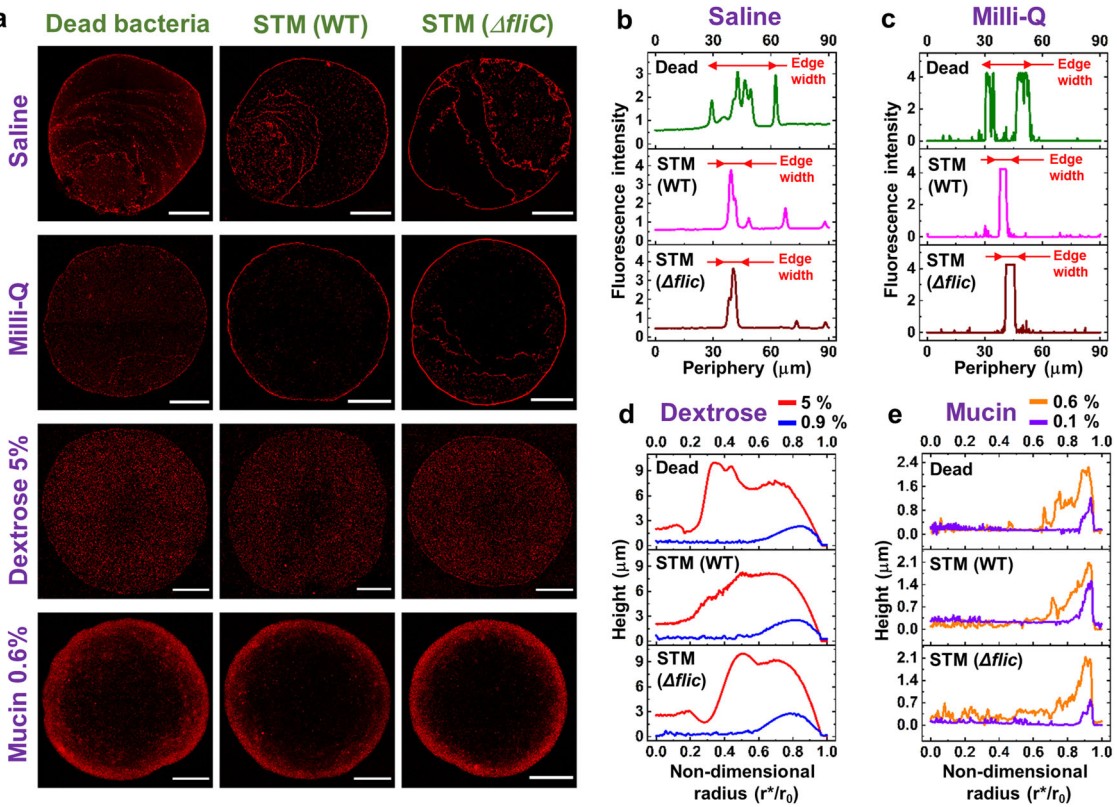

**Fig. 2 Precipitates of droplets containing *Salmonella* bacterial strains in different media. a** Confocal images (10× magnification) of the droplet precipitates for dead bacteria, STM (WT), and STM (Δ*fliC*) in saline, milli-Q, dextrose 5wt%, and mucin 0.6wt% media. Scale bars, 500 µm. **b** Fluorescence intensity at the edge for dead bacteria, STM (WT), and STM (Δ*fliC*) in saline medium (Supplementary Data 1). **c** Fluorescence intensity at the edge for dead bacteria, STM (WT) and STM (Δ*fliC*) in milli-Q medium (Supplementary Data 2). **d** Radial profilometry for dead bacteria, STM (WT) and STM (Δ*fliC*) in dextrose 5wt% and 0.9wt% media (Supplementary Data 3). **e** Radial profilometry for dead bacteria, STM (WT) and STM (Δ*fliC*) in mucin 0.6wt% and 0.1wt% media (Supplementary Data 4).

mucin medium. It is observed that the deposition pattern on these surfaces (Supplementary Fig. 2) are similar to the glass surface (control surface), used for further investigation in the current study. Furthermore, virulence study for all the surfaces are performed, as explained in the later sections.

Intra-medium bacterial strains are also found to govern the bacterial transport as the corresponding deposits differ considerably, particularly in neutral media (Fig. 2b, c). Therefore, to unfold the variation, we study the pattern formation in a more localized way, i.e., concentrating on the edge.

**Bacterial depositions at the edge**. Figure 3a, c, e, f show the confocal images in the edge region for all the media in ×63 magnification under confocal microscopy. We observe that WT and Δ*fliC* have narrower and aligned depositions, while in the case for dead, bacterial deposition is scattered over a larger region in which some differently oriented individual bacterium can be detected. The SEM images also corroborate similar understandings (Fig. 3b, d).

The variations in intra-medium bacterial depositions in case of nutrient-rich media are somewhat tricky to distinguish, as the bacteria are scattered throughout the contact area as well as near the edge of droplet. The increased width of solute deposition near the edge, as quantified from profilometry, is responsible for such dispersed bacterial deposition, further supported by the live cell videos (Supplementary Movies 2 and 3). In the case of mucin (Supplementary Movie 3), in the later stages of evaporation, we observe the formation of a thick polymer layer (final thick layer shown in SEM images in Fig. 3g) at the edge. Bacteria get entrapped

within the sticky polymer restricting their movement toward the edge, resulting in more diffused patterns. For dextrose, the bacterial agglomeration near the droplet pinned edge is rather interesting (Supplementary Movie 2). Initially, capillary flow transports the bacteria toward the edge. However, when the solvent depletion is about to be completed, dominant flow circulation is perceived at the final stages. This disturbs the bacterial arrangements near the edge and the bacteria is swept toward the center. It is plausible that in case of concentrated dextrose media, solutal marangoni forces may have triggered this flow circulation, thereby resulting in wide deposition of bacteria (evident in Fig. 2d), further investigated by transport dynamics.

Usually for all media, during initial stages, the evaporation of bacteria laden drops is governed by capillary flow; due to which the bacteria move toward the contact line of the drop and subsequently adhere to the surface. However, as the time progresses, particularly for the fluid mediums containing solutes (dextrose and mucin) of higher concentration, localized solute difference gives rise to surface tension gradient, thereby generating the solutal Marangoni flow. Both the flow regimes control the deposition and pattern dynamics of bacterial agglomeration. The above-mentioned factors in the nutrient-rich media hinder the bacterial edge depositions, resulting in varying orientations of individual bacterium, thus reducing any intra-medium contrast. Therefore, in this section, we conclude that the flow dynamics in the droplet play a major role in bacterial pattern formation. This is further elucidated in the following section, in which we analyze the flow dynamics using µ-PIV (micro-particle image velocimetry).

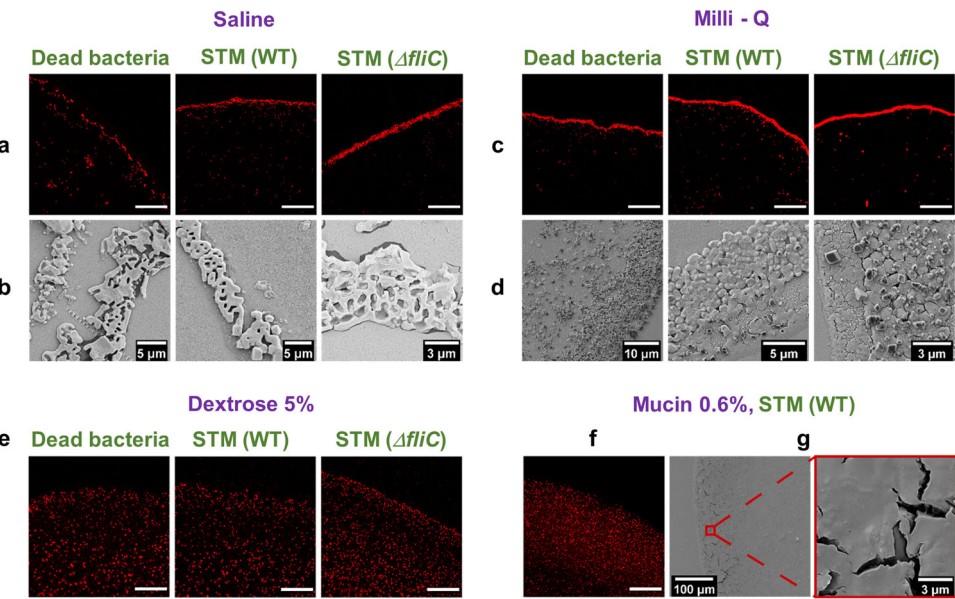

**Fig. 3 Bacterial deposition at the edge for four different media. a** Confocal (63× magnification) and **b** SEM images of the deposition at the edge for dead bacteria, STM (WT), and STM (Δ*fliC*) in saline medium. **c** Confocal (63× magnification) and **d** SEM images of the deposition at the edge for dead bacteria, STM (WT), and STM (Δ*fliC*) in milli-Q medium. **e** Confocal (63× magnification) images of the deposition at the edge for dead bacteria, STM (WT), and STM (Δ*fliC*) in dextrose 5wt% medium. **f** Confocal (63× magnification) and **g** SEM images of the deposition at the edge for STM (WT) in mucin 0.6wt% medium. Scale bars besides those specified, 50 μm.

**Transport dynamics of bacteria laden droplet.** Figure 4a–e illustrate the flow vectors (representing velocity of flow) for neutral and nutrient-rich media (experimental details are given in Methods section). The vector plots (Fig. 4a–d) are averaged over first 50 s of evaporation. Saline solution vector plot (Fig. 4a) depicts capillary dominated flow toward the edge, thus initiating the bacterial motion toward the periphery of the droplet forming a sharp edge. The bacterial deposition pattern and flow dynamics in milli-Q water are identical to saline solution. Therefore, to reduce redundancy, we have only analyzed saline solution as neutral medium. In case of mucin, the vector plots depicted in Fig. 4b (high and low concentration) show capillary flow which continues till the end.

For high concentration dextrose solution, vector plots correspond to two different planes of the droplet around the central region (Fig. 4d). We observe that at the upper plane ($h'/h_0 = 0.7$, where $h'$ is the height of the plane of interest from the base, and $h_0$ is the initial height of the droplet), flow vectors diverge from a central point, whereas, in the lower plane, it converges, thereby confirming marangoni flow[24]. In Fig. 4c, we observe a bifurcation region (the junction of opposite directional flow) in high concentration dextrose medium at an earlier stage compared to lower concentration counterpart, where only capillary flow persists. Moreover, in the final stage ($t*/t_f \geq 0.9$, where $t*$ is the time elapsed) of droplet evaporation (Fig. 4e), close to the edge, the low concentration medium starts to exhibit a bifurcation region. Contrarily, flow circulation is dominant in high concentration solution as the bifurcation region has already moved away from the edge toward the center. This depicts that in low concentration dextrose medium, the capillary flow is dominant over the weak marangoni convection, whereas, for concentrated media, subdued capillary flow is overcome by stronger marangoni current leading to graded distribution of bacterial deposits. Thus, marangoni flow governs the diffused edge pattern in dextrose medium. Marangoni flow has been studied by many researchers[25–27] considering the effects of temperature on the surface tension gradient. However, in the present study, we have not observed any marangoni flow in

non-solutal medium like milli-Q water which proves that the marangoni forces in the current study is concentration gradient dominant. Furthermore, Fig. 5a shows the velocity profiles, averaged over 400 μm of radial distance from the edge, comparing neutral and nutrient-rich media with time. The final evaporation time ($t_f$) is aligned for all cases (to account for different evaporation time for the media) (Fig. 5c) to compare and understand the dynamics for the solutions before complete precipitation. The saline and dextrose solutions have similar velocity profiles and the final surge in velocity is due to a rush hour effect[28]. However, in mucin (0.6 wt% concentration) solution, the velocity becomes high at an early stage owing to enhanced rush-hour effect. This is due to the formation of a polymeric layer that grows at a significant rate at the edge, reducing the effective contact angle. Moreover, the polymer separates out of water at the edge in the initial phase (corresponding time 0.25 $t_f$) due to evaporation[29], thus increasing the mass flux at the three-phase contact line. These factors aggravate the velocity of mucin in the region of interest. However, due to the polymer formation around the periphery, mucin transforms to sticky gel resulting in early velocity decay. The normalized decrease in contact angle due to droplet evaporation is shown in Fig. 5b for different media. For saline and dextrose base solution, $\theta_c/\theta_{c0}$ ($\theta_c$ is the instantaneous contact angle, $\theta_{c0}$ is the initial contact angle) is calculated until the droplet starts receding and the time of evaporation for each medium is illustrated in Fig. 5c. It is also noted that the higher velocities in mucin medium lead to a shorter evaporation time in comparison with saline and dextrose. It has been found that bacteria-laden drops also behave similarly with the respective base media as far as $\theta_c/\theta_{c0}$ decrement is concerned (Supplementary Fig. 3 (Supplementary Data 9–12)). Although the bacteria-laden dextrose drops exhibit similar characteristics compared to base fluid, no de-pinning is observed due to bacterial adherence to the solid surface near the three-phase contact line.

Further, we have developed mathematical models to map with the bacterial deposition patterns heuristically to rationalize the above-mentioned observations. Three different types of models

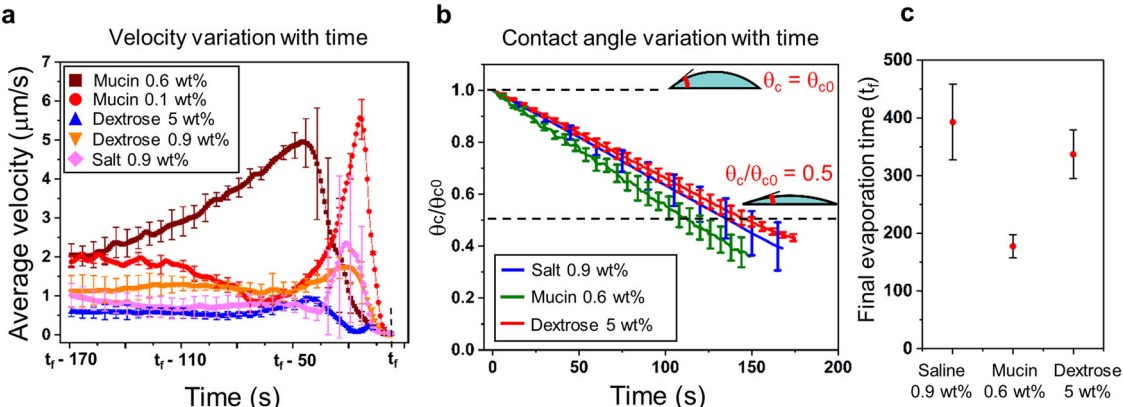

**Fig. 4 Flow visualization for saline, mucin, and dextrose medium droplets. a** Vector plot (averaged over the first 50 s) depicting capillary flow at the edge for a droplet of saline solution. **b** Vector plot (averaged over the first 50 s) depicting capillary flow at the edge for droplets of mucin 0.6 wt% and 0.1 wt% solutions. **c** Vector plot (averaged over the first 50 s) depicting weak solutal marangoni for dextrose 0.9 wt% and strong solutal marangoni for dextrose 5 wt% at the edge. **d** Vector plots (averaged over the first 50 s) depicting marangoni flow for dextrose 5 wt% at two different heights ($h'/h_0 = 0.1$ and 0.7) in the center of the droplet. **e** Receding flow due to marangoni at the final stages of evaporation for dextrose 0.9 wt% and 5 wt%. $t^*$ is the elapsed time. $t_f$ is the final time of evaporation.

**Fig. 5 Velocity and contact angle plots for saline, mucin, and dextrose medium droplets. a** Velocity profiles for saline and nutrient rich media at the later stages of evaporation. $t_f$ is aligned for all media for effective comparison (Supplementary Data 5). **b** Normalized contact angle variation ($\theta_c/\theta_{c0}$) with time for saline, mucin 0.6 wt%, and dextrose 5 wt% (Supplementary Data 6). **c** $t_f$ for saline, mucin 0.6 wt% and dextrose 5 wt%.

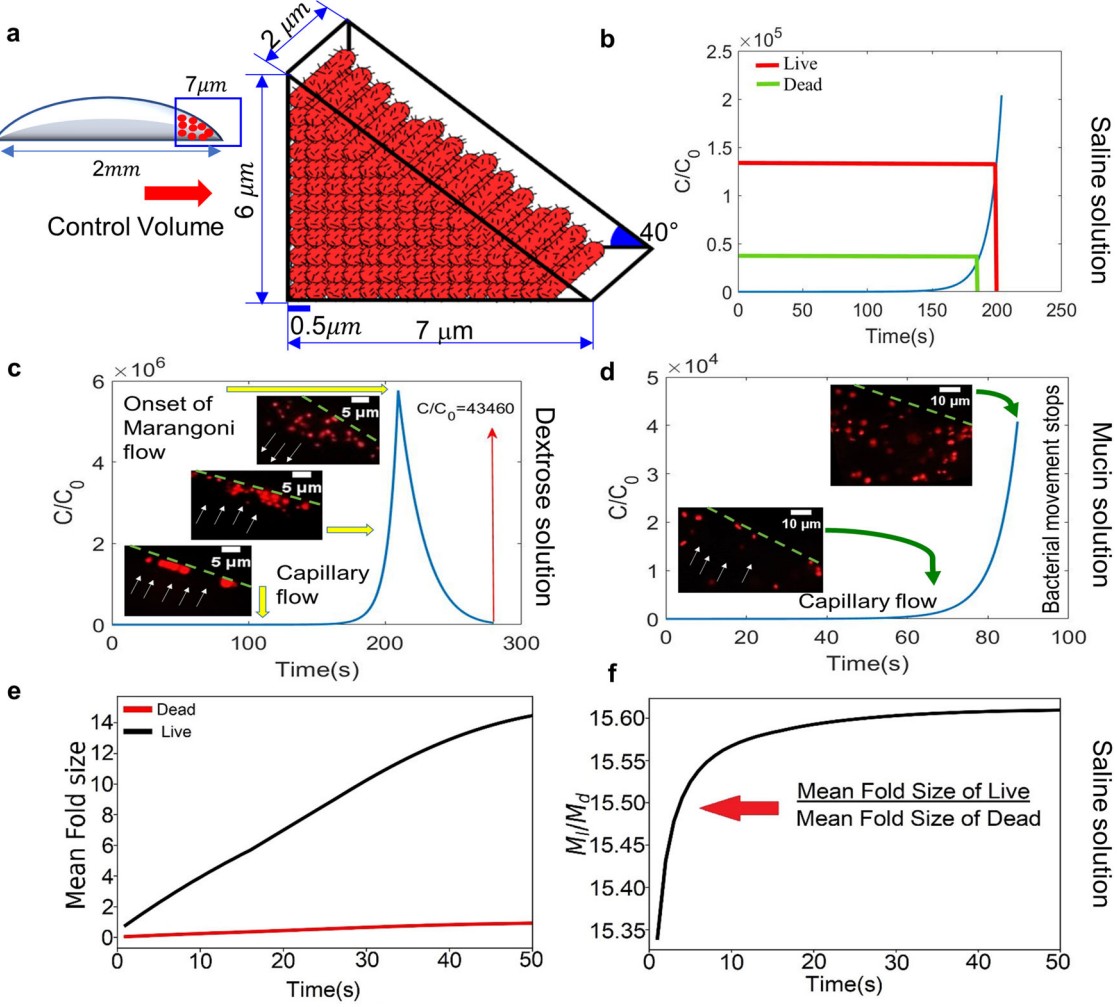

**Fig. 6 Modeling of bacterial deposition at the edge. a** Schematic of Model A depicting the increase in bacterial concentration in the control volume at the edge with time. **b** Plot of normalized concentration vs time in the control volume for saline solution. **c** Plot of normalized concentration vs time in the control volume for dextrose solution. **d** Plot of normalized concentration vs time in the control volume for mucin solution. **e** Plot of the number of mean fold formation vs time for dead and live bacteria in saline solution. **f** Ratio of mean fold size for live to dead along time.

are used to describe different perspectives of the patterns. In model A, we have computed the increase in bacterial concentration with time, explaining the reason for edge width differences in case of live (STM WT and STM Δ*fliC*) and dead bacteria. In model B, we have considered the aggregation dynamics of bacteria in an evaporating droplet, explaining the compact and scattered deposition patterns. Finally, in model C, we have used a more localized approach, focusing on the dynamics of bacteria from a microscopic perspective.

**Mathematical modeling**

*Model A: spatio-temporal variation in bacterial concentration.* To estimate the number of bacteria present in the edge of the droplet, we first develop a solid rod model as illustrated schematically (Fig. 6a). To characterize the edge deposition patterns universally across all media, we have considered the well-defined edge width of live bacteria for neutral solution as the computational length-scale in the radial direction. The initial averaged contact angle for all media is experimentally estimated to be around 40°. We placed each bacteria of size 2 × 0.5 μm dimension in the triangular section of the droplet edge, resulting in a 3D control volume (Fig. 6a) along the edge width of 7 μm. The total number of bacteria that arrange within the control volume considering tight packing in radial direction is estimated to be 86 CFU (colony

forming unit). Since PFA fixed dead bacteria becomes a rigid cell structure[30], assumption of individual bacterium as solid rod is justifiable. However, there are plenty of instances that show live bacteria can squeeze in stressed conditions and forms a spherical shape[31]. One of the most frequently observed behaviors in the nutrient starvation response of Gram-negative bacteria such as *Salmonella* Typhimurium is the size reduction and cell morphology conversion from rod to coccoid shape[32] as coccoid-shaped cells can survive for a very long time in the absence of nutrients. Therefore, in the same control volume, the maximum number of live bacteria that can get deposited increases three times (length of bacteria reduces due to coccoid shape) more than that for a solid rod. Hence the maximum bacterial deposition for dead bacteria in the control volume is 86 CFU and live bacteria is 258 CFU which can be used as the upper limit for increase in concentration with time at the edge for our mathematical model, as discussed below.

Transport of bacteria within the droplet is described by a modified Keller–Segel equation[33]:

$$\frac{\partial C}{\partial t^*} + \nabla.((u_l^* + u_{chx}^*)C) = \nabla.(D_{bl}\nabla C) \quad (1)$$

Where $C$ is the bacterial concentration ($CFU/m^3$), $u_l^*$ is the liquid velocity, $u_{chx}^*$ is the chemotactic velocity of bacteria toward

nutrient concentration gradient and $D_{bl}$ is the diffusivity of bacteria ($m^2/s$) within solution. Due to the dominant convective flow in an evaporating droplet, bacterial transport is completely dependent on fluid velocity (Figs. 4 and 5). Thus, bacterial chemotactic velocity is negligible in such small time-scale and hence neglected in the above Eq. (1). Moreover, as the flow dynamics in droplet evaporation is convection dominant, therefore by scaling analysis, diffusion term ($O(10^{-14})$) also becomes negligible in comparison to the transient term ($O(10^5)$ and advective term $O(10^2)$). Hence the modified equation in this model is given by:

$$\frac{\partial C}{\partial t^*} + \nabla.(u_l^* C) = 0 \qquad (2)$$

Here, velocity of fluid flow is considered to be dominant in radial direction and numerically calculated from ref. [34],

$$\begin{aligned} u_l = {} & \frac{3}{8}\frac{1}{1-t}\frac{1}{r}[(1-r^2) - (1-r^2)^{(-\lambda(\theta_c))}]\left(\frac{z^2}{h^2} - 2\frac{z}{h}\right) \\ & + \left\{\frac{rh_0^2 h}{r_0^2}(J\lambda(\theta_c)(1-r^2)^{(-\lambda(\theta_c)-1)} + 1)\left(\frac{z}{h} - \frac{3z^2}{2h^2}\right)\right\} \end{aligned} \qquad (3)$$

The change in contact angle $\theta_c$ with time is experimentally calculated from Fig. 5b. $u_l$, $t$, $r$, and $z$ are the non-dimensional liquid velocity, time, radial direction, and axial direction of the droplet respectively, $\lambda(\theta_c) = \frac{1}{2} - \frac{\theta_c}{\pi}$ is a parameter reflecting the uniformity of evaporation, $J$ is the evaporation flux along the droplet surface, $h$ is the non-dimensional height of the droplet, $r_0$ is the initial contact line radius. The velocity profile calculated from Eq. (3) is only used to determine velocity due to capillary flow and are in the same order (1–3 μm/s) with salt (0.9 wt%) velocity profile (Fig. 5a).

Figure 6b–d illustrate the transient variations of bacterial concentration near the edge in saline, dextrose and mucin media, respectively. An increase in bacterial concentration in the control volume with time is observed (Fig. 6b). The time required by dead bacteria to fully occupy the control volume is denoted by a green line and time required by live bacteria is denoted by red line. We observe from the plot that in ~180 s, dead bacteria completely occupy the control volume while live bacteria take ~200 s to occupy the edge width of 7 μm. The model dictates that dead bacteria deposition after 180 s should be outside the control volume explaining the wider edge deposition for dead bacteria. Figure 6c depicts an increase in bacterial concentration in the control volume at the initial phase due to capillary flow dominance but decreases eventually due to marangoni flow (shown in live cell images). An approximate temporal function of the marangoni velocity is computed from μ-PIV data manifesting flow reversal at the edge which forces the bacterial concentration to reduce within the control volume. The reduced concentration over the total evaporation time becomes lower than that of the maximum occupancy of dead bacteria ($C/C_0 = 4.5 \times 10^4$), eliminating the variation between dead and live bacteria in dextrose medium. An increasing profile of bacterial concentration in mucin solution is depicted in Fig. 6d due to capillary flow. However, bacterial movement in the edge region is restricted due to the sticky polymer formation, after approximately $t^* = 87$ s time scale calculated from μ-PIV data. Thus, the bacterial concentration in the control volume before the onset of velocity decay, is low enough to have any significant variation in live and dead bacterial deposition.

This model successfully demonstrates the bacterial deposition patterns (Figs. 2 and 3) concentrated at the edge. The rise in bacterial concentration in the control volume has the ability to enhance the agglomeration potential. Therefore, to understand the agglomeration dynamics of the bacteria at the periphery of the droplet, we look into the next model.

*Model B: agglomeration of bacteria.* The increase in bacterial concentration at the edge with time helps the bacteria to interact with each other and form aggregates. Bacterial behavior is usually the result of a competition between a random walk-based diffusion process and accumulation. Nevertheless, in this case, as convection flow is much more dominant than diffusion, the coagulation mechanism becomes more significant. When bacteria come in close proximity with each other due to the rise in concentration at the edge, factors responsible for two bacteria to attach in a convection-based dynamical system are Brownian motion and shear rate of the flow. This phenomenon is explained by an aggregation model given by Smoluchowski Equation[35],

$$\frac{dN_k}{dt} = \frac{1}{2}\sum_{i+j=k} k_{i,j}N_iN_j - N_k\sum_{j=1}^{\infty} k_{j,k}N_j, \qquad (4)$$

where $N_k$ gives the probability of bacterial number density to aggregate, $N_i$ gives the number of folds of bacteria interacting with another fold $N_j$. $k_{i,j}$ is the kernel operator that gives the rate at which one bacterium will interact with another bacterium. The kinetic constant for the shear-driven aggregation of bacteria depends on randomizing motion of Brownian diffusion and time-dependent shear rate $\gamma_s$. Increasing shear rate leads to an increase in the collision rate affecting the aggregation dynamics. The shear-induced kernel[36] is given by

$$k_{i,j} \simeq \sqrt{\frac{\mu\gamma_s a^3}{k_B T}}e^{\frac{6\pi\mu\gamma_s a^3}{k_B T}}, \qquad (5)$$

where $k_B$ is Boltzmann's constant, $T$ is the absolute temperature, and $\mu$ is the dynamic fluid viscosity (Supplementary Table 1), $a$ is the bacterial length, spatio-temporal shear rate $\gamma_s = \frac{U^*}{h^*}$, where $U^*$ is the velocity of the bacterial transport similar to the dimensional fluid velocity $u_l^*$ ($u_l = u_l^* \times t_f/r_0$) due to strong flow convection, calculated from Eq. (3), and $h^*$ is the instantaneous height of the droplet.

We solve the above equations numerically by Filbet and Laurençot Flux Method (FLFM) which was developed by Filbet and Laurençot[37]. The scheme is based on a finite volume method that tracks each fold's volume distribution with time. When one bacterium interacts with another, they aggregate to form two-fold of bacteria. Continuing in this manner (Supplementary Fig. 4), the volume distribution of one-fold bacteria reduces with time while volume distribution for a larger number of folds gradually increases with time. For a convergent numerical simulation, we approximated an upper limit for the growth in bacterial population. The maximum number of mean folds of the bacterial population density is considered to be approximately 14, equal to the count of live bacteria in neutral medium, positioned in the edge in a radial manner. Bacterial stress factor at the droplet's edge is taken into account, thus reducing the bacterial length $a$ in the case for live bacteria in neutral medium.

The volume distribution of mean folds for live (STM WT and STM $\Delta fliC$) and dead bacteria in saline medium (Fig. 6e) quantitatively predict the maximum number of mean folds that the bacteria can form at the edge in each case. The probability of forming folds in live bacteria is relatively high than its dead counterpart. Figure 6e depicts that live bacteria take 50 s to reach the upper limit of mean folds which is approximately 15 times larger than the mean fold developed by dead bacteria in that period (Fig. 6f). Thus, the mean folds explain bacteria's aggregation dynamics, depicting a dense and continuous pattern in live bacteria, whereas discontinuous and scattered patterns for dead. In dextrose medium, bacterial aggregates

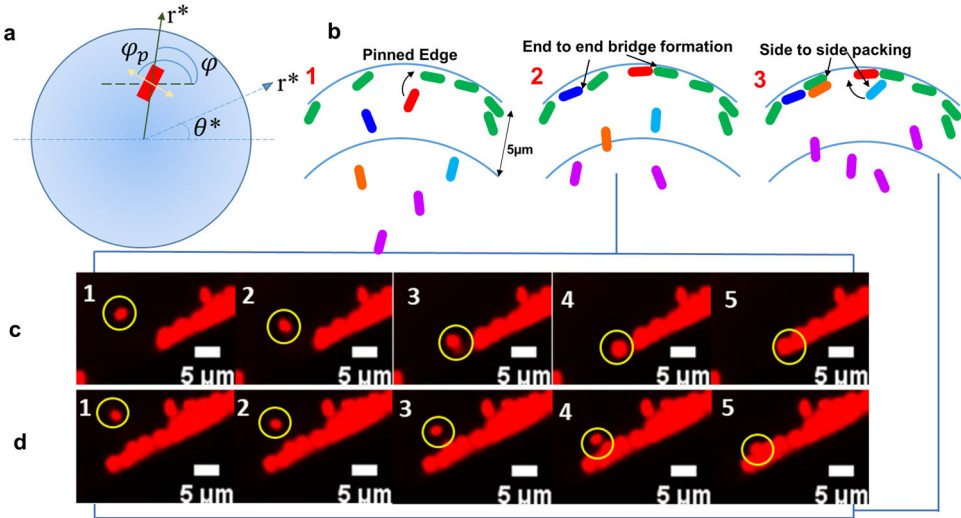

**Fig. 7 Model C Schematics and live Cell Images. a** Schematic of the top view of the droplet with polar coordinates ($r^*, \theta^*$) and of the particle-induced deformation, approximated as a quadrupole. The angle $\phi_p$ indicates the orientation of the quadrupolar rise axis, represented as a two-headed arrow. The angle $\phi$ indicates the orientation of the principal axis of curvature. **b** Schematic representation of Model C depicting bacterial orientation and packing having end-to-end and side-to side attraction. **c**, **d** Sequence of live cell images depicting bacterial end-to-end and side-to side attraction where, (**b(2)**) and (**b(3)**) are the schematics that represents (**c**) and (**d**), respectively. Bacterial orientation starts from 5 μm distance.

are formed in the initial phase but the flow reversal forces them to break down. The defragmentation of bacterial folds in dextrose medium results in scattered deposition. Moreover, in mucin medium, bacterial movement is restricted at the edge due to the sticky polymer formation. Therefore, the Smoluchowski equation does not hold for mucin medium. This elucidates the difference in bacterial pattern formation in a heuristic manner from a macroscopic point of view.

Now, to understand the pattern formations in a more localized approach, we have observed the droplet evaporation dynamics in live cell imaging under 63× magnification, focused at the edge. We observed that at the immediate periphery of the droplet, the bacterial cells orient themselves from radial direction to tangential direction which is parallel to the edge of the droplet. Moreover, the bacterial cells are observed to migrate toward other bacteria already deposited at the edge to form end-to-end orientation and side-to-side assembly. The orientation and packing phenomenon of the bacterial cells toward the edge are explained using a third model.

*Model C: bacterial tiling.* From a fluid dynamics perspective, the three-phase interface in droplet is a well-known region for directed migration and assembly of particles[38–43]. When particles migrate to the interface, spontaneous, long-range interactions occur owing to capillary energy, given by the product of the surface tension and the area of the distortion. When distortions induced by neighboring particles overlap, the interfacial area decreases, resulting in capillary interactions that cause particles to attract and assemble. This assembly occurs at random locations on the interface determined by sites of initial encounter between the particles. A similar phenomenon is observed with bacteria, getting deposited in droplet's edge, as shown in live cell images (Fig. 7c, d). Figure 7b provides an explanatory schematic to depict this bacterial interaction to form packing structures, evident from SEM images (Fig. 3b, d). Furthermore, end-to-end and side-to-side bacterial interaction has been observed while forming packing structure in which the change in bacterial orientation becomes significant within 5μm distance from the droplet's edge as obtained from live cell microscopy images (Fig. 7c, d). Bacteria interacting with pre-deposited bacteria at the edge, rotate and align coaxially with the latter, forming linear chains all oriented end-to-end.

Migration of bacteria toward the periphery of the droplet is induced by changes in interfacial energy given by ref. [44],

$$E = -\pi\gamma H_p R_p^2 \frac{1}{R_c(r^*)} \cos 2(\phi_p - \phi), \qquad (6)$$

where, $\gamma$ is the surface tension, $H_p$ is the deformation amplitude, i.e., fluid interface deformation around the particles, $\phi_p$ defines the orientation of the quadrupolar rise axis, $R_p$ is the bacterial radius, $R_c$ is the radius of curvature and $\phi$ is the angle defining the orientation of the principal axes of curvature at $r^*$ explained in the schematic geometry (Fig. 7a). This energy drives bacterial rotation, with a torque given by

$$\tau = -\frac{\partial E}{\partial \phi_p} \qquad (7)$$

comparable in magnitude to the capillary energy itself. Hence, to reduce the torque experienced by the bacterial body, they rotate along with the interface. Therefore, in the timescale of droplet evaporation $t_f \approx 3$–7 mins, the droplet dynamics supercedes any bacterial motility factors as the torque is experienced by both dead and live bacteria shown in the live cell imaging video (Supplementary Movie 4). The interfacial energy $E$ is directly proportional to fluid's surface tension $\gamma$ and deformation amplitude $H_p$. The values of interfacial surface tension are experimentally obtained within the range of 70–72 mN/m for all the media (Supplementary Table 2). However, the difference between the bacterial deposition patterns for live and dead bacteria is very prominent, particularly for neutral media. The fact is for continuous elevation in bacterial concentration and less fluid at the edge[20], live bacteria experience stress and start deforming. Additionally, STM WT and STM ΔfliC, have gliding motility[45] which increases the deformation amplitude. On the other hand, PFA-fixed dead bacteria have a rigid cell structure that restricts their deformation. Therefore,

$$E \propto H_p \qquad (8)$$

which makes the interaction potential in live bacteria greater than that of dead bacteria depicting the lattice-like structure in live bacteria at the edge (Fig. 3b, d).

From the above-mentioned observations and mathematical modeling, we predict that, in the small-time scale (minutes) of an evaporating droplet, deposition patterns are fluid dynamics dominant. Moreover, as there is no difference in pattern formation between STM WT and STM Δ*fliC*, it is apt to infer that flagella in STM WT do not play any significant role in the precipitation dynamics. The mathematical models were intended to qualitatively assess the trend of experimental observations. The models are not quantitatively compared with experiments, rather gives a holistic idea on the flow pattern of an evaporating sessile droplet containing dead and live bacteria. Moreover, we have reported that the velocity for dextrose and saline are of same order while mucin has higher velocity (Fig. 5a). We have analyzed the temporal data of droplet evaporation dynamics in presence and absence of bacteria and observed that there is hardly any variation in the contact line dynamics of the drop. Therefore, the estimated strain rate (calculated based on the height variation of droplet and capillary flow velocity) for dextrose and saline fall in similar range, while in case of mucin it is one order higher. The viscosity of mucin (0.6 wt%) is approximately than 1.5 times high than that of saline (0.9 wt%) and dextrose (0.9 wt%). The evaporation driven flow generates shear stress in the fluid which is experienced by the live bacteria. The flow shear stress exerted toward the edge of the droplet in the final 170 sec of evaporation is calculated from the velocity (Fig. 5a) and viscosity (Supplementary Table 1) for different media. The shear stress exerted on the bacteria in saline solution is 4.08 µPa, dextrose (5 wt%) is 3.38 µPa and mucin (0.6 wt%) is 31.96 µPa. The shear stress of mucin is estimated to be one order high than both dextrose and saline. This mechanical stress and low moisture starvation situation is predicted to succour live bacteria squeeze for longer survivability even though viability is reduced significantly. However, the authors currently can only hypothesise due to limited information regarding how flow shear stresses affects bacterial cells and are translated to direct biological responses. This can open future prospect of research considering different kinds of bacteria and environment. However, to calculate the survival time span and infectivity of STM in different medium's dried fomite, we have performed infectivity test explained in the following section.

**Viable *Salmonella* in fomite can cause enhanced infection in mouse.** In previous sections, we have conjectured that the bacteria deform itself during starvation in the evaporated droplet. To investigate the role of starvation within evaporated droplet on bacterial survival, we decided to enumerate the viable bacterial count in the sessile droplets obtained from saline (Fig. 8a), and milli-Q water (Fig. 8b) samples 5 h after evaporation. We have found that the viability of wild type and *fliC* deficient strains of *Salmonella* Typhimurium reduces significantly after evaporation of the solvent. We found that the viable bacterial count of wild type and *fliC* mutant strains of *Salmonella* dropped from $10^5$ CFU to $10^2$ CFU in 0.5 µl of saline suspension 5 h post evaporation. This suggests the presence of multiple stress factors in dried droplets like lack of nutrients or starvation stress, low moisture or water depletion stress etc. However, the excessively reduced bacterial count of *fliC* deficient *Salmonella* (from $10^5$ CFU to $10^1$ CFU) in dried deposition of milli-Q (Fig. 8b) which lacks any nutrients, suggests that the ability of wild type *Salmonella* to thrive in the stressed environment is better than *fliC* mutant.

Surprisingly, in comparison with the evaporated droplets of saline and milli-Q, the higher bacterial burden for both STM (WT) and Δ*fliC* (~$10^3$ CFU of bacteria) in 5 h old dried droplets of high (0.6 wt%) and low (0.1 wt%) concentration mucin (Blue bars; Fig. 8c, d) suggests that the nutrient abundance of fomites

plays an extremely important role to control the viability of the bacteria. The mucin Muc2 of the intestinal mucus layer is associated with the restriction of *Salmonella* infection in the mouse model[46]. On the contrary, transmembrane mucin MUC1 aggravates the invasion of epithelial cells by *Salmonella* through the apical route[47]. The better survival of STM (WT) and Δ*fliC* in mucin solution can be attributed to its heavily glycosylated proteinaceous nature, which can act as a source of nutrients and energy required for the nourishment of bacteria in the dried droplets[48]. Evaporated saline, and milli-Q droplets were more restrictive in allowing the survival of STM (WT) and Δ*fliC* than mucin. To further ascertain the role of nutrient abundance of dried sessile droplets in the survival of bacteria, we used varying concentration of mucin (high, 0.6 wt%; and low, 0.1 wt%; Fig. 8c, d) and dextrose (high, 5 wt%; and low, 0.9 wt%; Fig. 8e, f) and checked the bacterial viability at 5h, 24 h, 48 h, and 72 h post evaporation. Compared with the pre-evaporated sample, we have found a consistent and significant reduction in the viable bacterial count for both the strains with time ($t^* = 5$ h, 24 h, 48 h, and 72 h) in the evaporated droplets of mucin and dextrose. However, until 72 h post evaporation, viable bacteria (both STM WT and Δ*fliC*) were found for both high and low mucin concentrations (Fig. 8c, d). Contrary to this, our data suggest that the evaporated droplets of dextrose, which is not nutritionally enriched like mucin, can restrict the growth of bacteria[49]. We have found that STM (WT) can survive in the dried droplets of high concentration of dextrose until 72 h post evaporation where the estimated viability dropped from $10^5$ CFU to $10^3$ CFU at 5th hour post-evaporation and remained constant till 48th hour after which it further dropped below $10^1$ CFU (Fig. 8e). In line with this observation, STM Δ*fliC* was found to be struggling severely to survive even in a high dose of dextrose where viable bacterial count was not found beyond 5 h (Fig. 8e). The inability of *fliC* mutant and wild type strains of *Salmonella* to grow in dried depositions of low concentration of dextrose beyond 5th and 72nd hours strongly supports the role of nutrient availability on bacterial survival in dried droplets.

We further decided to investigate whether the formation of fomite compromises the virulence of wild type *Salmonella*. Our data suggest that the wild type *Salmonella* present in fomite precipitates of saline, mucin, and dextrose are more prone to be phagocytosed by RAW264.7 cells (Fig. 9a). The phagocytosis of wild type bacteria obtained from dextrose, mucin, and saline fomites were found to be almost 3 times higher than non-evaporated planktonic culture. We have also found that in murine macrophages the intracellular fold proliferation of wild type bacteria retrieved from dried depositions of mucin is 3 times higher than the liquid phase bacterial culture. The hyper proliferation of wild type *Salmonella* obtained from dried deposition of mucin in RAW264.7 macrophages suggests a significant increase in the virulence of the bacteria after evaporation. In line with this observation, the comparable fold proliferation of the bacteria from the evaporated precipitate of saline and dextrose in comparison with their liquid phase counterpart (Fig. 9b) states that the formation of fomite after the evaporation of solvent does not make wild type bacteria virulence deficient. The probable reason behind higher entry of dried bacteria in murine macrophages is accounted to the mechanical and low-moisture stress experienced by the bacteria. Moreover, the flow shear stress close to the edge for mucin medium is 5 times larger than saline and dextrose, probably causing hyper-proliferation of STM WT in macrophages.

After entering the macrophages wild type *Salmonella* resides in a modified membrane-bound compartment called *Salmonella* containing vacuole (SCV), which is essential for its successful intracellular survival and pathogenesis[50]. The co-localization of LAMP-1 (which is a standard SCV marker) with wild type

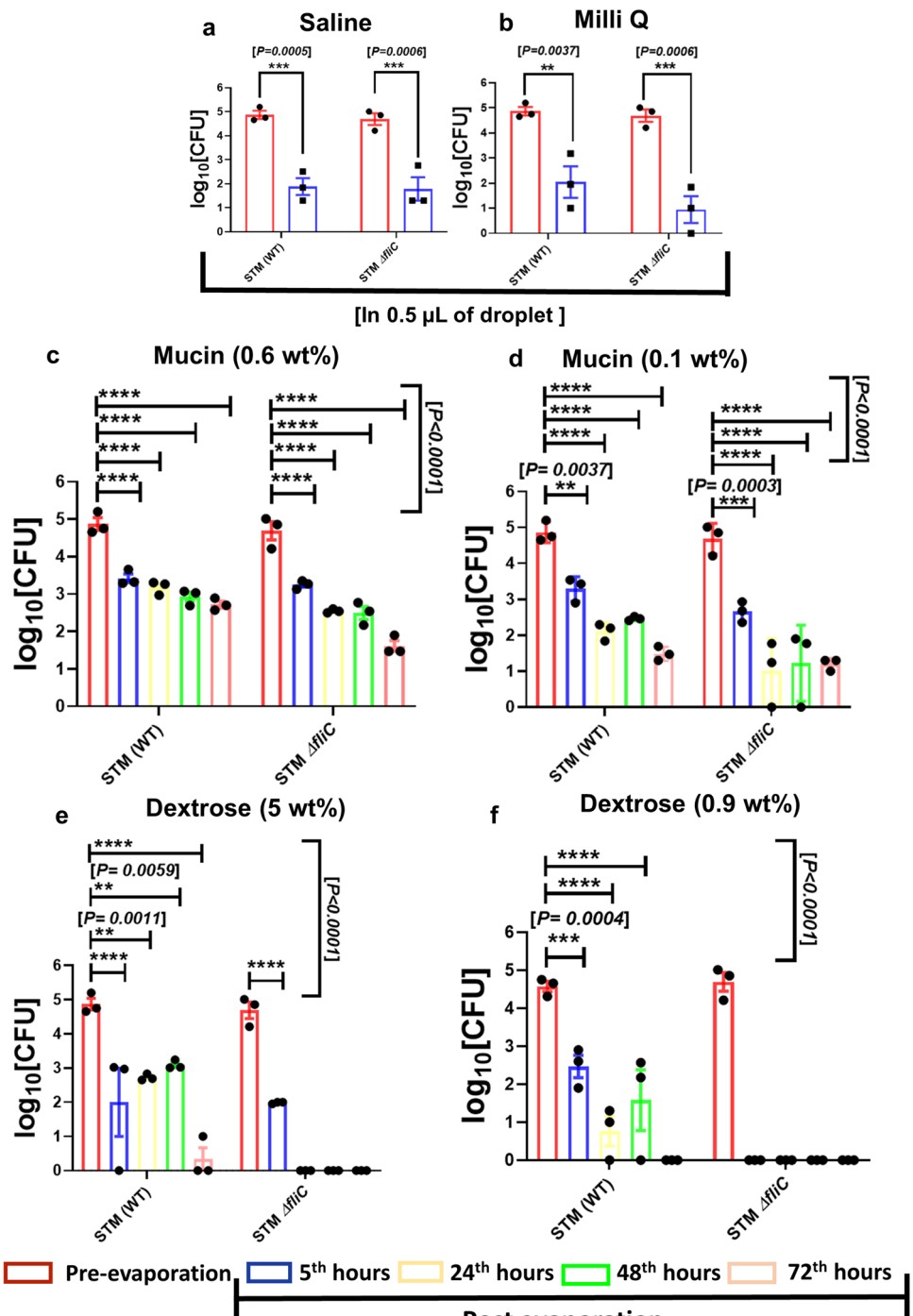

**Fig. 8 The viability of wild type and *fliC* deficient strains of *Salmonella* is significantly compromised post-evaporation (Supplementary Data 7).** The assessment of in vitro viability of STM (WT) and Δ*fliC* present in 0.5 µL droplets of **a** saline (0.9% W/V), **b** milli-Q water before and 5 h after evaporation, (*N* = 3, *N* = biological replicates, mean ± SEM). The time dependent viability of STM (WT) and Δ*fliC* present in 0.5 µL droplets of high and low concentration of **c** mucin (0.6 wt% for high concentration) and **d** mucin (0.1 wt% for low concentration) and **e** dextrose (5 wt% for high concentration) and **f** dextrose (0.9 wt% for low concentrations) solutions before and 5 h, 24 h, 48 h, and 72 h post evaporation, (*N* = 3, biological replicates, mean ± SEM). In every independently done experiment a single bacterial culture (either wild type or *fliC* mutant) was used to prepare a common pre-evaporating sample, which was followed by drop casting of the bacteria in saline, milli-Q, mucin, and dextrose solutions. (*P*)* < 0.05, (*P*)** < 0.005, (*P*)*** < 0.0005, (*P*) **** < 0.0001, ns = non-significant, (2way ANOVA).

*Salmonella* obtained from dextrose, mucin and saline fomite precipitates suggests the vacuolar confinement of the bacteria in macrophages (Fig. 9c). In human body macrophages can control the propagation of invading bacterial pathogens by several mechanisms. The generation of reactive oxygen species (ROS), reactive nitrogen intermediates (RNI), cationic antimicrobial peptides by macrophages during early stage of infection can restrict the proliferation of bacteria. The fate of wild type *Salmonella* Typhimurium inside macrophages depends upon its entrapment into SCV. The acidic environment of SCV

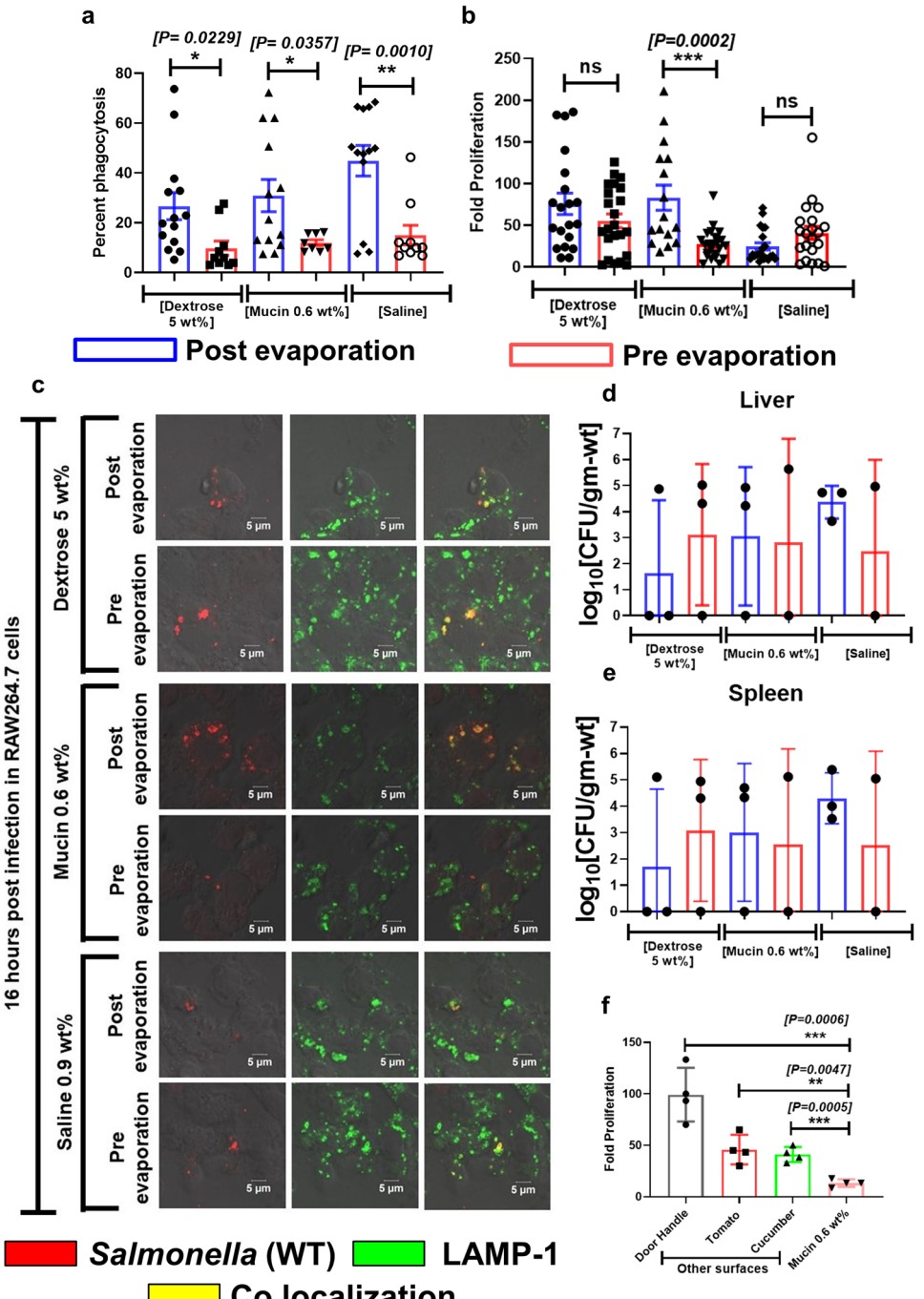

**Fig. 9 Viable *Salmonella* Typhimurium present in fomite can cause infection in mouse model (Supplementary Data 8). a** Percent phagocytosis of STM (WT) from post evaporated (fomite) and pre evaporated (non-fomite) samples by RAW264.7 cells, ($n = 4$, $N = 2$, $N$ = biological replicates, mean ± SEM). Fomite and non-fomite samples of *S.* Typhimurium were prepared from dextrose (5%), mucin (0.6 wt%), and saline (0.9 wt%) solutions respectively. **b** Comparison between the intracellular fold proliferation of post evaporated and pre evaporated STM (WT) in RAW264.7 cells, ($n = 4$, $N = 2$, $N$ = biological replicates, mean ± SEM). **c** Representative images depicting the investigation of the vacuolar niche of fomite and non-fomite wild type *Salmonella* in RAW264.7 cells 16 h post infection by LAMP-1 staining, ($n = 10$, $N = 2$, $N$ = biological replicates). 16 hours post infection the cells were fixed with 3.5% paraformaldehyde. The cells were stained with appropriate primary and secondary antibodies (details in "Methods" section). The bacterial burden in the **d** liver and **e** spleen of 3–4 weeks old BALB/c mice orally gavaged with fomite and non-fomite wild type *S.* Typhimurium, ($n = 3$, $n$ = number of mice per groups). **f** Comparison between the intracellular fold proliferation of post evaporated and pre evaporated STM (WT) obtained from mucin solution (0.6 wt%) in RAW264.7 cells. The bacterial deposition was collected from the surface of door handle, tomato, and cucumber, respectively, ($n = 4$, $n$ = number of sets of macrophages infected from each surface, mean ± SD). $(P)^* < 0.05$, $(P)^{**} < 0.005$, $(P)^{***} < 0.0005$, $(P)^{****} < 0.0001$, ns = non-significant, Student's $t$ test.

triggers the expression of SPI-2 encoded virulent factors which are required for creating a successfully replicative niche of *Salmonella* in macrophages[51]. The wild type *Salmonella* reconstituted from dried droplets of mucin, saline and dextrose were

found to be staying inside SCV during late phase of infection in murine macrophages, which can also be considered as one of the reasons behind their successful intracellular growth. Taken together, it can be explicitly inferred that the survival of wild

type *Salmonella* in dried droplets or fomites depends upon the availability of nutrients.

Dried bacterial droplets on fomites reduced its viability with time. The presence of nutrients in the dried droplets (in this study- mucin of 0.6 wt% concentration) can maintain a fixed but relatively small population of bacteria with time. However, this reduction in the viability of the bacteria hardly hampers its virulence as elucidated by in vitro cell culture and in vivo mouse model infection assays. We further checked the in vivo pathogenesis of wild type *Salmonella* obtained from fomite laden droplet precipitates of dextrose, mucin, and saline in BALB/c mouse model. We found that the wild type *Salmonella* recovered from the 5 h old dried droplets of dextrose, mucin and saline are capable of causing infection in mouse. The comparable burden of wild type *S.* Typhimurium in the liver (Fig. 9d) and spleens (Fig. 9e) of mice infected with fomite and non-fomite liquid phase bacteria strongly supported the fact that deposition of bacteria on fomite does not compromise the virulence of the pathogen. We further tested the potential of wild type *Salmonella* isolated from the mucin depositions on realistic surfaces commonly found, namely door handle, skins of tomato, and cucumber to infect RAW264.7 cells (Fig. 9f). We found that in all three cases, wild type *Salmonella* can infect and proliferate well in macrophages. The intracellular fold proliferation of the bacteria isolated from organic surfaces is almost three times greater than that of the planktonic culture, having similar range of proliferation with glass slides. On the other hand, in case of bacteria isolated from the surfaces of steel door handle showed more than five fold increase in their intracellular proliferation, corroborating a higher risk of disease transmission.

## Discussion

Evaporation-flow coupled dynamics of microscale sessile droplets containing PFA fixed dead (STM (WT)), and different strains of *Salmonella* (STM (WT) and STM (Δ*fliC*)) on fomites are studied in neutral and nutrient-rich media. The dried precipitates are further investigated for subsequent infection rates.

Evaporating sessile droplet on a hydrophilic surface has the three-phase contact angle always between zero to 90°[52]. As droplet evaporation starts, droplet with contact angle less than 90°, undergoes constant contact area mode of evaporation. In this mode, the solid-liquid contact area remains constant, whereas the CA gradually decreases as shown in Fig. 5b[53]. As the evaporation occurs from the three-phase contact line (TCL), the droplet's edge evaporates at a higher rate than in the center, so the fluid particles must flow radially outward to the boundary to compensate for the faster evaporation of the thin liquid layer. This radial outward flow occurring inside a droplet is known as Deegan flow[54–58]. In this particular flow regime, the droplet usually maintains a constant radius and the contact angle decreases. The deposition of insoluble particles immersed in the fluid at the periphery creates the coffee-ring effect (Fig. 2a). Besides, multiple ring patterns may form based on the particle-size selection mechanism near the TCL[59] or the phase separation phenomenon of salt crystallization in an evaporating droplet of solutions of a polyelectrolyte and salt[60]. The other flow regime is the Marangoni flow (Fig. 4d), which can be the opposite of the Deegan flow and is driven by surface tension gradient[61–64]. In general, either temperature or concentration gradient or both can cause change in surface tension which gives rise to Marangoni forces. In the present study, we believe that the concentration gradient might have played dominant role in Marangoni flow. However, it is quite likely that, local temperature differences may also lead to Marangoni convection within an evaporating droplet.

The post-evaporated precipitates exhibit distinct pattern formation across the bacterial strains and media due to the flow dynamics. For neutral media, there is a capillary-dominated flow during evaporation that carries the bacteria toward the droplet's periphery. STM (WT) and STM (Δ*fliC*) have a dense and continuous edge, while dead bacteria have a wider and discontinuous edge. Similar bacterial deposition patterns at the droplet's edge have also been observed in[65–75] for an evaporating droplet. For the nutrient-rich media, the bacterial deposition is observed to be significantly diffused at the edge. In mucin, it is due to the formation of a thick polymer layer, observed using SEM imaging, which restricts the flow of the bacteria. Interestingly for dextrose, the solutal marangoni effect, observed using micro-PIV, reverses the flow near the edge, thus disrupting the pattern formation. The observed patterns are heuristically explained by mathematical modeling considering the squeezing of live bacteria into spherical shapes during stressed conditions, in contrast with dead bacteria with a rigid cell structure. The model also quantitatively predicts that dead bacteria have a lower chance of forming aggregates than STM (WT) and STM (Δ*fliC*), thus explaining the discontinuous bacterial deposition patterns at the edge for neutral media.

Further localizing our scope, a focused flow imaging near the droplet contact line depicts bacterial approach toward the edge causes deformation of the fluid interface. This deformation leads to dynamic directional changes resulting in end-to-end and side-to-side bacterial assembly at the perimeter. The rotation of bacteria to reduce the torque experienced by their body is observed for both live and dead bacteria predicting fluid dynamics dominance. The fluid interface deformation around STM (WT) and STM (Δ*fliC*) is reasoned to be significantly larger than dead bacteria due to its ability of gliding motility and to deform from rod to coccoid shape for flow and low moisture stress conditions. The modeling predicted that the deformation amplitude proportionally increases the interfacial energy, leading to a much larger interaction potential for STM (WT) and STM (Δ*fliC*) and thus a more compact structure at the edge in comparison with dead bacteria. The experimental observations and mathematical models also clearly indicate that the bacteria's flagella do not play any significant role in the patterns formed within the short time-scale of evaporation in our study.

The comparative viability of these post-evaporated bacteria for different media is studied and observed that viability of STM WT reduces up to three orders for neutral solutions whereas two orders for dextrose (5 wt%) and one order for mucin (0.6 wt%). Moreover, for nutrient-rich media, bacteria is found to survive for multiple days. Interestingly, though viability is reduced by orders, post-evaporated STM (WT) bacteria shows an increase in percent phagocytosis depicting a higher entry percentage inside the RAW264.7 cells. Furthermore, the fold proliferation is comparable for saline and dextrose media and significantly higher for mucin. This is caused due to five times higher flow shear stress in mucin medium than saline and dextrose endured by the bacteria during evaporation, bringing about this enhanced infectivity through fomites. This proves that the increased bacterial virulence is governed by the flow stress rather than just a function of nutrition of the media. The augmented virulence is a previously unseen outcome from fluid dynamics-based interaction of bacteria in sessile droplets on different substrates. The potency of the fomites in causing an infection is confirmed from observations of the bacterial load in the liver and spleen of BALB/c mice orally gavaged with bacteria collected from dried precipitate on fomites. Hyper-proliferation of bacterial burden is also detected from commonly found surfaces such as steel door handle, surfaces of tomato and cucumber, confirming enhanced virulence of post-evaporated bacteria. Though the study has been conducted using

*Salmonella* Typhimurium but we conjecture that all types of bacteria may exhibit similar results.

## Methods

**Preparation of bacterial suspension.** Single colonies of STM (WT): RFP and STM Δ*fliC*: RFP were inoculated in 5 mL of LB broth in the presence of either ampicillin (concentration, 50 μg/mL for wildtype strain of *Salmonella*) or kanamycin and ampicillin together (concentration, 50 μg/mL for *fliC* knockout strain of *Salmonella*) and incubated overnight in a shaking incubator at 37 °C at 170 rpm. 1.5 mL of the overnight grown stationary phase culture was centrifuged at 6000 rpm for 10 min, and the pellet was washed once with double autoclaved sterile Milli-Q water. Finally, the pellet was resuspended in 500 μL of autoclaved Milli-Q water. The prepared bacterial suspension was subjected to serial dilution by five folds thrice in saline solution (0.9 wt%), dextrose solution (0.9 wt% and 5 wt%), mucin solution (0.6 wt% and 0.1 wt%) and autoclaved Milli-Q water respectively. 0.5 μL of this diluted suspension was used for dropcast on sterile glass slides, which was followed by imaging with confocal microscopy and viability assay. We have found that 0.5 μL of diluted suspension corresponds to 105 CFU of bacteria, sufficient to cause infection in the human host[76,77]. To prepare the dead bacterial suspension, 1.5 mL of overnight grown STM (WT): RFP culture was centrifuged at 6000 rpm for 10 min and washed with sterile Milli Q water, as mentioned before. The pellet was resuspended in 4% paraformaldehyde (PFA) and incubated for 1 h. After the fixation was done, the dead bacteria were washed and resuspended in 500 μL of sterile Milli Q water. To enhance the dead bacteria's fluorescence intensity, propidium iodide (final concentration, 0.04 μg/μL) was added. The propidium iodide added dead bacterial cells were subjected to serial dilution in Milli Q water, saline, dextrose, and mucin solutions, as mentioned earlier for further studies.

**Experimental setup for contact line (CL) dynamics & fluorescence imaging.** Plain glass slides (purchased from Blue Star©) were kept in propan-2-ol bath and sonicated for 2 min, followed by rinsing with Kimwipes (Kimberly Clark International). A droplet volume of $0.5 \pm 0.1$ μl was gently placed on to the substrate (glass slide) using a micropipette (procured from Thermo Scientific Finnpipette, range: 0.2–2 μl) and was allowed to dry under controlled environment [temperature was maintained about 25–28 °C and relative humidity around 44–48% (measured with a sensor TSP-01, provided by Thorlabs)].

During the evaporation of the droplet, the contact line (CL) dynamics was observed from the top, using an optical microscope (Olympus). The light source in-line with the objective lens of the microscope illuminated the drop and the CCD camera (Nikon D7200), mounted on top of the microscope captured the images (30 fps) of the evaporating drop. Another digital camera (Nikon D5600) fitted with a zoom lens assembly (Navitar) recorded the shadowgraph images (1/5 fps) from side view with the help of a controllable LED light source (5W, purchased from Holmarc). The diffuser plate placed between light source and droplet was used for proper illumination. Image analysis were performed with freely available software ImageJ. To evaluate the dynamic behaviors of the RFP tagged bacteria within the evaporating droplet, the droplets were volumetrically illuminated using a mercury source with a filter for excitation at 532 nm wavelength. The emission signal from bacteria was captured for flow visualization using a PCO camera (CMOS) (captured at 2.14 fps) attached to the microscope at high magnifications (50× & 100×).

The droplet radius ($r_d$), estimated from image analysis was ~0.9 mm. Since this length scale is less than the capillary length ($l_c$) ($r_d < l_c = \sqrt{\frac{\sigma}{\rho g}} \approx 2.7$ mm, $\sigma \approx$ 70−72 mN/m [Supplementary Table 2] for all the media), the spherical cap assumption at the start of experiments remains valid for present cases, in agreement with other reports[24]. The initial contact angle of the droplet on glass was observed to be $40 \pm 5°$.

**μ-PIV for flow studies.** To understand the flow patterns inside different media, 2-D micro-particle image velocimetry (μ-PIV) was performed. The illumination was provided with a Nd:Yag laser (NanoPIV, Litron Laser) and the camera attached to a Flowmaster MITAS microscope [field of view (FOV): $600 \times 450$ μm, depth of field: 7 μm] captured the images. Neutrally buoyant red fluorescent particles (Fluro-Max, size: $860 \pm 5$ nm, procured from Sigma-Aldrich) were also added to the base solutions to trace the flow inside the evaporating droplet in absence of any bacteria. The images were acquired using a single frame-single pulse technique, at 1 fps and postprocessed using LaVision DaVis 8.4 software.

**Studying the edge deposition of the bacteria using live-cell imaging.** The edge deposition of the STM (WT): RFP, Δ*fliC*: RFP, PFA fixed dead bacteria resuspended in saline, mucin, and sugar solution was studied using Leica SP8 confocal microscope. Briefly, bacteria laden drops were put on the sterile rectangular glass coverslips. After placing the droplet on the coverslip, the samples were illuminated with a metal halide lamp. The edge deposition of bacteria was captured by live videography under ×63 magnification till the drop evaporated entirely. The standard laboratory conditions (temperature, 25–28 °C and relative humidity 44–48%) were maintained while capturing the droplet dynamics.

**Multiphoton laser scanning confocal microscopy.** The evaporated bacterial samples were analyzed by Zeiss LSM 880 NLO upright multi-photon confocal microscope using ×10, ×40, and ×63 magnification. The background corrections were done using base solutions (Milli-Q, saline, dextrose, and mucin solutions without bacteria). The Z stacks images were obtained using ZEN Black software (Carl Zeiss) to study the bacterial arrangements in the dried precipitates.

**Scanning electron microscopy & profilometry.** The dried precipitate patterns are viewed under SEM to observe the bacterial agglomeration and optical profilometer (Talysurf Hobson) was also used to quantify the thickness of the remnant precipitates.

**Calculation of bacterial viability in the evaporated samples.** Five hours post evaporation, the dried bacterial residues were rehydrated with 100 μL of PBS (pH = 7.4) and plated on *Salmonella Shigella* (SS) Agar. After 16 h of incubation at 37 °C temperature, the colonies that appeared on the plates were counted to estimate the number of viable bacteria. The pre-inoculum was also plated on SS agar to enumerate the number of bacteria present in 0.5 μL volume of suspension. In the time scale-dependent study of bacterial viability, the evaporated samples were rehydrated with PBS on the 5th, 24th, 48th, and 72nd hours and plated on SS agar enumeration viable bacteria. The $\log_{10}$ values of the CFU/mL obtained from plating were plotted with GraphPad Prism software (version 8).

**Intracellular cellular proliferation assay of bacteria.** 1 to $1.5 \times 10^5 RAW264.7$ cells (murine macrophages) were seeded in each well of a 24 well plate and infected with wild-type *Salmonella* Typhimurium either in liquid phase or dried in mucin, dextrose, and saline solutions, respectively. The infected cells were incubated in 5% $CO_2$ at 37 °C for 30 min and washed with sterile PBS to remove extracellular bacteria. To inhibit the growth of extracellular bacteria, the cells were treated with 100 μg/mL of gentamycin for an hour and 25 μg/mL of gentamycin till the end of the study. The cells were lysed with 0.1% of triton X 100 at 2 h and 16 h post-infection. The lysates were plated on *Salmonella Shigella* agar to enumerate the number of bacteria that have entered and proliferated within the cells. The fold proliferation and percent phagocytosis of the bacteria was calculated using the following formulae

Fold proliferation = [CFU at 16 h]/ [CFU at 2 h]

Percent phagocytosis = [CFU at 2 h]/ [CFU of pre-inoculum]

**Preparing samples for confocal microscopy.** 1 to $1.5 \times 10^5 RAW264.7$ cells were seeded on the top of sterile glass coverslips and infected with wildtype *Salmonella* Typhimurium either in liquid phase or dried in mucin, dextrose, and saline solutions, respectively. The infected cells were incubated in 5% $CO_2$ at 37 °C for 30 min and washed with sterile PBS to remove extracellular bacteria. To inhibit the growth of extracellular bacteria, the cells were treated with 100 μg/mL of gentamycin for an hour and 25 μg/mL of gentamycin till the end of the study. 16 h post infection the cells were fixed with 3.5% paraformaldehyde. The cells were stained with rat anti-mouse LAMP-1 (1:100) and rabbit anti-*Salmonella* (1:100) primary antibodies for an hour in room temperature which was further followed by the treatment of the cells with anti-rat dylight 488 (1:100) and anti-rabbit cy3 (1:100) secondary antibodies. The coverslips were mounted with anti-fade reagent and the edges were sealed with transparent nail paint on glass slide for image acquisition. The images of infected macrophage were taken by Zeiss LSM 880 NLO upright multi-photon confocal microscope using ×63 oil immersion lens. The images were processed using ZEN Black software (Carl Zeiss) to study the bacterial localization within macrophage.

**Animal infection studies.** Total six cohorts of 3–4 weeks old BALB/c mice were orally gavaged with either 5 h old fomite (three cohorts) or liquid phase non-fomite (three cohorts) wildtype *Salmonella* Typhimurium. On 5th day post infection, the infected mice were sacrificed and dissected in sterile condition. Liver and spleens were isolated from the infected mice and homogenized with sterile glass beads. The lysate was serially diluted and plated on SS agar for estimating the bacterial burden in these two organs. All the animal experiments were performed after receiving the approval from Institutional Animal Ethics Committee. The Guidelines given by National Animal Care were strictly followed throughout the study. (Registration No: 48/1999/CPCSEA).

**Statistics and reproducibility.** The CL evaporation of bacteria laden droplets were carried out at-least five times for each cases to ensure repeatability of the data. The PIV experiments of a given sample were also conducted three times to estimate the average flow velocity. The time-dependent and independent in vitro viability assay to estimate the survival of bacteria in fomites have been performed thrice independently. The percent phagocytosis and the intracellular survival assay of the bacteria have been performed twice independently with four technical replicates. The animal experiment has been done once with sample size of 3 mice/group. The experiment to estimate the intracellular viability of wild type *Salmonella* from organic and inanimate surface has been done once with four technical replicates. For the biological experiments, to analyze the data, 2way ANOVA and two-tailed

Student's *t* test have been performed wherever applicable. The mean with standard error mean (for the experiments done independently with technical replicates) and standard deviation (for the experiments done once with technical replicates) was determined with the help of GraphPad Prism 8.4.3 (686) software.

**Reporting summary**. Further information on research design is available in the Nature Research Reporting Summary linked to this article.

## Data availability

All data are available in the main text or the Supplementary materials. All materials are available from the corresponding author upon genuine request.

## Code availability

The codes are available from the corresponding author upon reasonable request.

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

## Acknowledgements

We thank Durbar Roy for mathematical modeling assistance and Omkar Hegde for helping in capturing images through scanning electron microscope and making optical profilometry measurements. We acknowledge MNCF (a characterization center in CENSE at IISc) for allowing access to SEM and optical profilometry equipment. We also acknowledge the esteemed reviewers for their valuable suggestion. We thank the central animal facility of Indian Indian Institute of Science for proving us BALB/c mice to conduct our experiments. S.B.: DRDO Chair Professorship, D.C.: DAE-SRC fellowship, ASTRA- Chair fellowship, TATA Innovation grant, DBT-IOE partnership grant.

## Author contributions

Conceptualization: S.B. and D.C. Methodology: S.B., D.C., A.R.C., and S.M. Investigation: A.R.C., S.M., R.P., A.C., and A.A. Visualization: R.P., A.R.C., S.M., and A.C. Funding acquisition: D.C. and S.B. Project administration: D.C. and S.B. Supervision: D.C. and S.B. Writing— original draft: S.M. and A.R.C. Writing: editing and revision: S.B., D.C., A.R.C., S.M., R.P., and A.C.

## Competing interests

The authors declare no competing interests.
