## [Peer Review File · Communications Biology]

Reviewers' comments:

Reviewer #1 (Remarks to the Author):

The manuscript titled "Spatiotemporal evaporating droplet dynamics on fomites enhances long term bacterial pathogenesis" shows that bacteria dried on fomites can become viable even after experiencing the stresses during the evaporative solvent loss. This article explains the evaporative deposition of bacteria at the drop edges and demonstrates the consequences using animal study. I believe that it is an important study, which demonstrates the pathogenesis after drying and rehydration. The results have importance especially in the context of the current pandemic scenario. Therefore, I recommend this article for publication in Communications Biology after considering the questions and suggestions.

1) Authors show that the bacterial deposition at the edges depends on the solutal Marangoni flow. However, Bacteria have the tendency to adhere to the surface near the contact line. They also have mutual interaction. These factors also affect their deposit pattern and dynamics. Authors may point to this aspect.

2) The authors claim that the increased bacterial virulence is governed by the flow stress rather than just a function of the nutrition of the media. Where is the data showing the relationship between shear stress and viability? There can be a correlation between the two aspects. However, there should be data or previous studies demonstrating that shear stress increases viability. Also, explain why should the shear increase the viability? Any previous studies?

3) Are the results optimized with roughly one evaporation time? If not what is the effect of the rate of evaporation on the viability of the bacteria? If the evaporation is fast (say 3-7 min as written in the manuscript), do the bacteria get enough time to express the mechanisms (for example, potassium uptake) to cope with increasing salt/chemical stress?

4) Salmonella is a water-borne pathogen, so how aerosol and fomite-induced virulence going to impact its transmission is not clear. Authors can comment about it in the manuscript.

5) In figure 8, the title is that the viability of the WT is compromised. But the same figure also shows the same scenario for the mutant type as well. It was a bit confusing. So maybe the figure caption title can be modified.

Minor points

- 1) The section on the biological study should be separated into different paragraphs.
- 2) Some typos

Reviewer #2 (Remarks to the Author)

The manuscript reports flow dynamics and deposition patterns in a bacteria laden droplet evaporating on various surfaces representing fomites. A systematic experimental study was carried out to study the transport and deposition of Salmonella Typhimurium (STM) in nutrient rich as well as neutral medium during evaporation on a flat surface. Finally, the authors have presented mathematical models for bacterial transport and agglomeration. The results reported in the manuscript are new and useful. However, the manuscript in the present form needs some changes, in particular the explanation of results from the angle of fundamental concepts that has been established through several past studies on evaporating sessile droplets. It is also desired that the results from the model predictions should have been at least qualitatively compared with the experiments. The phenomena of flow reversal is interesting and should have been discussed in context with the works reported in literature. I would also like the authors to clarify the following observations:

Pp4: Why the edge deposition should be influenced by the concentration profile?

Pp4: was there any study carried out on rough surfaces?

Pp4: it is surprising to note that the deposition is scattered over a large region for dead bacteria!

Should not the coffee ring effect give edge deposit even for dead bacteria?

(Fig. 4, pp5): The solutal Marangoni need not necessarily cause flow reversal. There are recent works that report flow reversal at low contact angle (Ristenpart et al, PRL, 2007; Zhang et al, Pre 2014; Thockchom et al., Microgravity Sc. Tech., 2021). The velocity vectors plotted in Fig. 4 show flow reversal but does it match with the predictions of these studies? What was the contact angle at which the authors observed the flow reversal?

Pp7: I fail to understand why there should be so large difference in the estimate of CFU values for live and dead bacteria? How was it arrived at?

Pp7: How can you neglect chemotactic velocity based on small time scale since it depends on nutrient concentration gradient and there could be regions of very high chemotactic velocity!

Pp8: The evaporation flux can be very different for bacteria laden droplet and therefore, the estimate of radial velocity (Eqn 3) will change. It has been found in many studies that bacteria tend to accumulate at the air-water interface.

Pp11: If the deposition patterns are fluid dynamics dominant then the shear stress values should scale with the viscosity of the solution (saline/dextrose/mucin). Did the authors observe this scaling?

Fig. 2A: Why the deposition pattern for saline is highly asymmetric compared to other fluid medium?

Fig. 5: One would expect faster evaporation time for saline compared to Mucin and Dextrose considering their high viscosity. Why this is not the case here? Was the humidity maintained the same in all the cases?

Fig. 6: The assumption of bacteria stacked in a regular fashion is too simplistic. In reality they would be randomly deposited at the edge and therefore should not the CFU values in the model will be unrealistic!

Answer to Reviewer's queries

Ref: - COMMSBIO-21-1615-T

Manuscript entitled "Spatiotemporal evaporating droplet dynamics on fomites enhances long term bacterial pathogenesis"

Reviewer 1:

The authors thank the reviewer for appreciating the work and for the insightful comments.

Comments:

I believe that it is an important study, which demonstrates the pathogenesis after drying and rehydration. The results have importance especially in the context of the current pandemic scenario. Therefore, I recommend this article for publication in Communications Biology after considering the questions and suggestions.

Answer:

The authors thank the reviewer for recommending the current paper for publication in Communications Biology. The questions and comments are addressed and the revised version is amended incorporating the suggestions.

Questions

1. Authors show that the bacterial deposition at the edges depends on the solutal Marangoni flow. However, Bacteria have the tendency to adhere to the surface near the contact line. They also have mutual interaction. These factors also affect their deposit pattern and dynamics. Authors may point to this aspect.

Answer:

The authors agree to the point mentioned by the reviewer. Usually for all media, during initial stages, the evaporation of bacteria laden drops is governed by capillary flow; due to which the bacteria move towards the contact line of the drop and subsequently adhere to the surface. However, as the time progresses, particularly for the fluid mediums containing solute (dextrose and mucin) of higher concentration, localized solute difference gives rise to surface tension gradient, thereby generating the Marangoni flow. Both the flow regimes control the deposition and pattern dynamics of bacterial agglomeration.

As recommended by the reviewer this point is incorporated in the revised version.

2. The authors claim that the increased bacterial virulence is governed by the flow stress rather than just a function of the nutrition of the media. Where is the data showing the relationship between shear stress and viability? There can be a correlation between the two aspects. However, there should be data or previous studies demonstrating that shear stress increases viability. Also, explain why should the shear increase the viability? Any previous studies?

Answer:

The following table shows that the fluid shear stress and viability of bacteria may have reasonable dependency. One can observe that viability of mucin is maximum among the three as the corresponding shear stress estimated over the entire droplet lifetime is also highest. In case of saline and dextrose, the respective shear stresses and bacterial viabilities are of similar order, one order less compared to mucin. In practice, viability may depend on multiple factors, such as selection of bacteria, type of mutants, the ambient atmosphere conditions (temperature, relative humidity) and the host liquid medium to name a few. In this study, we can observe from Fig. 8 that, even for the same liquid medium, the viability differs between WT and Δ fliC; thereby indicating that survival response may also vary based on type of mutations. Further, the choice of droplet medium (mucin, dextrose, and saline) also determines the viability of a given bacteria. Therefore, expressing bacterial viability only in terms of shear stress may be a premature effort as we should not ignore the contribution of several other variables in regulating the bacterial viability.

Solution	Shear Stress (μPa)	Viability (CFU)
Saline	4.08	$O(10^2)$
Dextrose	3.38	$O(10^2)$
Mucin	31.96	$O(10^3)$

The authors did not claim that shear stress increases viability. As shown in Fig. 8, it can be clearly observed that viability of the system reduces with time. However, the post-evaporation virulence data (Figs. 9A and 9B) show that in all three mediums, wild type Salmonella can infect and proliferate well in macrophages. Besides, the Figs. 9D and 9F corroborate similar findings. The authors presume that this augmented virulence is due to the combined role of fluid shear and presence of nutrients in the host medium. Nickerson et al. [1] suggested that mechanical and physical signals sensed by the microbe may be integrated with other environmental signals and transduced into a biochemical response. In this case, the shear stress integrated with nutrition of the solution may have caused such response. Moreover, the fluid shear rate as shown in the table is very low which is caused by the droplet evaporation. Nickerson et al. [1] reviewed that low flow shear stress used in LSMMG (low shear modeled microgravity) enhances *Salmonella enterica* serovar Typhimurium virulence. However, the authors currently can only hypothesize due to limited information regarding how flow shear stresses affect bacterial cells and are translated to direct biological responses. This can open future prospect of research considering different kinds of bacteria and environment.

3. Are the results optimized with roughly one evaporation time? If not, what is the effect of the rate of evaporation on the viability of the bacteria? If the evaporation is fast (say 3-7 min as written in the manuscript), do the bacteria get enough time to express the mechanisms (for example, potassium uptake) to cope with increasing salt/chemical stress?

Answer:

The evaporation rate of a given bacteria laden drop is highly dependent on the host liquid medium (as can be noted from Fig. 5C). Among all the droplet mediums, evaporation of mucin is fastest, followed by dextrose and saline solution. While designing the bacterial viability experiments, the dried precipitates were collected after a particular interval (for example, 5h, 24 h etc.) for all the mediums and further relevant procedures were followed.

As far as the uptake is concerned, we have used solute concentration such that it is isotonic. The uptake of certain elements is observed primarily when the suspension medium is hypertonic. However, in the present case, this does not come into effect.

4. *Salmonella* is a water-borne pathogen, so how aerosol and fomite-induced virulence going to impact its transmission is not clear. Authors can comment about it in the manuscript.

Answer:

Figure: Schematic diagram of water borne route of *Salmonella* to cause infection.

The irrigation involves supply of water for producing the crops and vegetables. During water supply, droplets and aerosols of water can form, which can settle on different parts (such as leaf, stem etc.) of a given plant. Assume that the water is already contaminated with *Salmonella*. Now this microbe can adhere to plant leaves, and even on end product (for example, tomato [2]). This pathogen can transmit to humans and animals if the raw product is consumed without further processing. With time, as we observe that their virulence remains unaffected post drying of liquid medium (say, in case of mucin), these bacteria can further cause infection (Fig. S1 and Fig. 9F). In California belt of USA, many cases of such *Salmonella* outbreak have been reported.

In fact, there are plenty of articles that point out the transmission route of *Salmonella* to different species. This part is already included in the literature survey portion of the main manuscript. Despite being a gut pathogen and having a fixed orofecal transmission route via contaminated food and water, many studies have demonstrated the aerosol transmission of specific serovars of *Salmonella* Enterica such as *Salmonella* Typhimurium, *Salmonella* Agona, etc. in poultry animals [3, 4]. Aerosol mediated transmission of *Salmonella* Typhimurium happens from contaminated feces and that can further contaminate poultry birds after 2 to 4 hours of exposure [5]. Thus, settling of the bacteria on fodder and other surfaces leads to infection through ingestion causing food poisoning in poultry animals [6]. *Salmonella* Enteritidis, one of the leading causes of salmonellosis in poultry birds, can survive on the outer shell of contaminated eggs at a low temperature and low relative humidity in the presence or absence of any nutrients [7, 8]. The improper sterilization of the

medical equipment which is already contaminated with bacterial fomites is one of the major reasons behind healthcare-associated outbreaks of *Salmonella* [9]. The ability of *Salmonella* Typhimurium to survive on inanimate surfaces such as polypropylene, formica, stainless steel, and wooden surfaces in the presence or absence of the protein source at room temperature, 6 hours post-inoculation enhances the risk of contamination of consumable food items from the contact surface [10]. All these factors contribute to the significance of study of *S. Typhimurium* in droplets settled on fomites.

5. In figure 8, the title is that the viability of the WT is compromised. But the same figure also shows the same scenario for the mutant type as well. It was a bit confusing. So maybe the figure caption title can be modified.

Answer:

The modified caption of Figure 8: **The viability of wildtype and *fliC* deficient strains of *Salmonella* is significantly compromised post-evaporation.** The revised version of the manuscript also incorporates this change and is highlighted.

Minor points

1. The section on the biological study should be separated into different paragraphs.

Answer:

As suggested by the reviewer, the biological portion of the main text is divided into separated paragraphs by maintaining continuity.

2. Some typos

Answer:

The manuscript has been thoroughly checked and all the necessary steps (correction of grammatical and punctuation errors) have been undertaken to improve the article.

Reviewer 2:

The authors thank Prof. Anugrah Singh for appreciating the work and for the insightful comments.

Comments:

The manuscript in the present form needs some changes, in particular the explanation of results from the angle of fundamental concepts that has been established through several past studies on evaporating sessile droplets. It is also desired that the results from the model predictions should have been at least qualitatively compared with the experiments. The phenomena of flow reversal is interesting and should have been discussed in context with the works reported in literature.

Answer:

The results are explained from the angle of fundamental concepts that has been established through several past studies [11-23] on evaporating sessile droplets and are highlighted in the manuscript. The manuscript is amended in the Discussion section paragraph 2.

The mathematical models were intended to qualitatively assess the trend of experimental observations. The models heuristically explained the difference between deposition patterns of dead and live bacteria. The model selection was kept very simple in order to predict the differences observed in Fig 2 and 3 without considering actual complicated motion of bacteria including flagellar rotation and tumbling. Moreover, the CFU calculation given in model A is the upper limit of bacterial deposition that can happen in the control volume considered. The calculated CFU was fully based on geometrical analysis which may not be the only parameter in case of bacteria. Therefore, the models are not quantitatively compared with experiments rather gives a holistic idea on the flow pattern of an evaporating sessile droplet containing dead and live bacteria. The current paper is mainly focused on experimental studies of bacterial deposition pattern and pathogenesis.

The flow reversal phenomenon is discussed in the manuscript in context with the works reported in literature.

The changes are incorporated in the manuscript and are highlighted.

Questions

1. (pp4) Why the edge deposition should be influenced by the concentration profile?

Answer:

Generally, edge deposition of non-volatile solute and bacteria present in the droplet is influenced by capillary flow due to droplet evaporation. However, as mentioned in the

manuscript, if we compare high and low concentration of dextrose and mucin, we observed thicker edge profile for higher concentration counterparts (Figs. 2D and 2E). In case of high concentration dextrose, strong Marangoni forces diffuses the edge deposition, making it broader. Similarly, high concentration mucin produces higher polymer deposition in which bacteria get stuck, unable to move forward to the edge, thus extending the edge deposition. Therefore, solute concentration profile in the droplet influences the edge deposition patterns.

2. (pp4) Was there any study carried out on rough surfaces?

Answer:

All the experiments of different droplet media were initially conducted on glass slides. The details of slide preparation are included in the Material and methods section. The contact angle for all the media is approximately 40°. To understand the interaction with real surfaces, we have used tomato and cucumber skins along with door handle (steel). The mean value of roughness (R_a) of glass and steel, calculated from profilometry data, are 0.011 μm and 0.12 μm [24], respectively. We have not carried out any experiment on textured surfaces.

3. (pp4) It is surprising to note that the deposition is scattered over a large region for dead bacteria! Should not the coffee ring effect give edge deposit even for dead bacteria?

Answer:

- The authors clarify that coffee ring effect is observed for all the bacterial strains (WT, dead and *fliC* mutant), irrespective of the choice of media.
- Coffee rings are the result of capillary flow, due to which the undissolved solute/particle move towards the droplet periphery and settle.
- Dead bacteria have rigid structures [25], which will maintain its rod shape.
- The aspect ratio (ratio of length to breadth) of *Salmonella typhimurium* comes roughly about 3.5-4.
- While live bacteria are elastic, deformable and squeeze to coccoid shape under stress [26], for which the aspect ratio would decrease.
- Yunker et al. [27] had investigated the effect on shape dependencies of particles on deposition patterns. They have reported that suspensions with higher aspect ratios (> 3) exhibit a uniform deposit, rather than a dense coffee ring.
- This is the reason, why we observe, uniform and scattered coffee ring deposition for dead bacteria than live bacteria.

4. (Fig. 4, pp5) The solutal Marangoni need not necessarily cause flow reversal. There are recent works that report flow reversal at low contact angle (Ristenpart et al, PRL, 2007; Zhang et al, Pre 2014; Thockchom et al., Microgravity Sc. Tech., 2021). The velocity vectors plotted in Fig. 4 show flow reversal, but does it match with the predictions of these studies? What was the contact angle at which the authors observed the flow reversal?

Answer:

The authors agree with the reviewer that solutal Marangoni need not necessarily cause flow reversal. The three studies that the reviewer cites in this question consider the effects of temperature on the surface tension gradient at low contact angle and are all concerning single component droplets on a substrate and hence differ from our study as it does not include solutal Marangoni. Zhang et al, Pre 2014 [28] numerically determined that flow reverses at a

CA of 14 deg while Ristenpart et al, PRL 2007 [29] theoretically determines it to be at 31 deg. In Thokchom et al, Microgravity Sc. Tech., 2021[30] flow reverses at 34 deg, although the experimental parameters (initial contact angle 73 deg, droplet confined between parallel plates) significantly differ from our own and therefore is not quantitatively comparable.

For our study, the CA for dextrose 5 wt% at 90% of evaporation time ($t^*/t_r^* = 0.9$) is found to be 10.1 ± 3.1 deg. We use the CA of dextrose 5 wt% here with the knowledge that the CA for all mediums are similar as seen in Fig. S3 of supplementary material. The CA at the final stage of evaporation during which flow reversal occurs is comparable to the studies mentioned above. However, in the current study if thermal gradient has any effect on the surface tension change, then it would be observed for all media, including milli Q water. But the authors have not observed any such flow characteristics. Therefore, the authors conclude that solutal concentration effect dominates the Marangoni currents than thermal gradients in the current study.

5. (Pp7) I fail to understand why there should be so large difference in the estimate of CFU values for live and dead bacteria? How was it arrived at?

Answer:

- In the 3D control volume considered in Fig. 6A, by geometrical analysis, the maximum number of bacteria that can arrange within the control volume considering tight packing in radial direction is estimated to be 86 CFU (colony forming unit).
- The bacterial stacking is a simple model that is assumed as the velocity of bacteria is negligible as compared to the flow velocity.
- It is considered that bacteria simply follow the capillary flow of the fluid towards the interface due to evaporation.
- PFA fixed dead bacteria becomes a rigid cell structure [25], assumption of individual bacterium as solid rod is justifiable.
- Live bacteria can squeeze in stressed conditions and forms a spherical shape [26,31].
- In the same control volume, the upper limit of live bacteria that can get deposited increases three times (length of live bacteria reduces as now the shape of bacteria is considered a sphere) more than that for a dead bacterium (solid rod).
- Hence the maximum bacterial deposition for dead bacteria in the control volume is 86 CFU and live bacteria is 258 CFU which can be used as the upper limit for increase in concentration with time at the edge.

6. (Pp7) How can you neglect chemotactic velocity based on small time scale since it depends on nutrient concentration gradient and there could be regions of very high chemotactic velocity!

Answer:

A semisolid soft agar (0.15-0.5% w/v) medium is required to study the bacterial chemotaxis, where a constant gradient of nutrients is maintained to motivate the bacteria to move from a region of low nutrient density (where it has already consumed the nutrients) to an area of high nutrient density [32]. On the contrary, in our study, we have considered a homogeneously spread nutrient solution such as Dextrose and Mucin. In a homogenous system, the nutrient is spread across equally, and there is no gradient spread. Thokchom et al [33] observed that as sugar crystal kept at the center of the substrate is dissolved in the fluid, no more nutrition gradient occurs, and chemotaxis also stops.

Furthermore, sensing the nutrient gradient of the surrounding media by the bacterial chemoreceptors and transmitting the signal to the flagellar motor is a time-consuming process. Therefore, the timescale used to measure bacterial chemotaxis is in several hours [31], whereas the current study time scale is only 3-7 mins. The bacteria were also homogeneously spread over the media of the droplet such that no pre-accumulation of bacteria in the droplet could occur. So, chemotaxis of bacteria in the current experimental setup if exists will be very weak and momentary to have any effect on the current study, thus neglected.

Moreover, we have not observed any additional flow of bacteria in micro-PIV or particle tracking results. The velocity scale of a particular medium is same with and without bacteria.

7. Pp8: The evaporation flux can be very different for bacteria laden droplet and therefore, the estimate of radial velocity (Eqn 3) will change. It has been found in many studies that bacteria tend to accumulate at the air-water interface.

Answer:

In general, many researchers have point out that the evaporation flux is a strong function of temporal change in contact angle (CA). As noted from Fig. S3 (supplementary), for a given medium, there is minimum variation among CAs of different bacteria laden solutions with the parent medium (in absence of bacteria) case. Therefore, we infer that the difference in evaporation flux would be insignificant that justifies the use of eqn. 3 to estimate the radial velocity.

8. (Pp11) If the deposition patterns are fluid dynamics dominant then the shear stress values should scale with the viscosity of the solution (saline/dextrose/mucin). Did the authors observe this scaling?

Answer:

We have reported that the velocity for dextrose and saline are of same order while mucin has higher velocity (Fig. 5A). We have analyzed the temporal data of droplet evaporation dynamics in presence and absence of bacteria and observed that there is hardly any variation in the contact line dynamics of the drop. Therefore, the estimated strain rate (calculated based on the height variation of droplet and capillary flow velocity) for dextrose and saline fall in similar range, while in case of mucin it is one order higher. The viscosity of mucin (0.6 wt%) is approximately than 1.5 times high than that of saline (0.9 wt%) and dextrose (0.9 wt%). The shear stress of mucin is estimated to be one order high than both dextrose and saline.

9. (Fig. 2A) Why the deposition pattern for saline is highly asymmetric compared to other fluid medium?

Answer:

In case of saline droplet, after a certain time period ($t/t_f > 0.5$, t instantaneous time and t_f the total time taken for evaporation) during evaporation, the contact line (CL) starts receding from its original pinned locations. This phenomenon of contact line depinning is random and therefore, we see such asymmetric deposition patterns. Further, as the CL de-pins, formation of salt crystals (during the final stages of drying) is noted to be around a particular location, where the droplet CL remains pinned. However, in case of nutrient rich media (dextrose and

mucin), the CL remains pinned to the original contact area. Therefore, due to absence of any receding of CL, we observe deposition patterns are globally symmetric.

10. (Fig. 5) One would expect faster evaporation time for saline compared to Mucin and Dextrose considering their high viscosity. Why this is not the case here? Was the humidity maintained the same in all the cases?

Answer:

The viscosities of saline and dextrose (0.9 and 5 wt%) are almost identical as seen in Table S2 of Supplementary Material. The values are also provided in the table below. Further the velocity profiles with time are also similar for saline and dextrose as seen in Figure 5A of the manuscript. Therefore, the total evaporation time for saline and dextrose is comparable and in fact overlaps as seen in Figure 5C of the manuscript.

The viscosity of mucin 0.6 wt% is not significantly higher than the other mediums as seen in the table below, while for mucin 0.1 wt% it is similar to saline and dextrose. In the case of mucin, as explained on page 6 of the manuscript, there is a formation of a polymeric layer, and as the polymer separates out of the liquid it aggravates the velocities (see Figure 5A of the manuscript for velocity profile of mucin (0.1 and 0.6 wt%) with time). This results in a shorter evaporation time in comparison with saline and dextrose as seen in Figure 5C of the manuscript. The manuscript has been amended on Page '10' paragraph '1' to include the information that the higher velocities in mucin medium lead to a shorter evaporation time in comparison with saline and dextrose.

The humidity was maintained at 44 - 48% for all cases as stated in the Methods section.

Fluid medium (w/o bacteria)	Viscosity (mPa.s)
Saline (0.9 wt%)	1.2
Dextrose (5 wt%)	1.3
Dextrose (0.9 wt%)	1.2
Mucin (0.6 wt%)	1.7
Mucin (0.1 wt%)	1.3

11. (Fig. 6) The assumption of bacteria stacked in a regular fashion is too simplistic. In reality they would be randomly deposited at the edge and therefore should not the CFU values in the model will be unrealistic!

Answer:

The authors agree that the assumption of bacterial stacking in a regular fashion is a simple model to heuristically understand the pattern formation observed from the experiment. From micro-PIV results and bacterial velocity tracking, we have concluded that the velocity of bacteria is negligible as compared to the flow velocity. Bacteria simply follows the capillary flow of the fluid towards the interface due to evaporation. Therefore, the assumption of bacterial stacking is justified. The bacterial deposition in reality will be random, so we randomly chose any such control volume where we considered the stacking model. In that control volume at the edge, the CFU values given are the upper bounds of bacterial deposition that can form. The initial bacterial concentration is fixed from which the initial CFU of the solution in that given volume is calculated and the increase in concentration is thus plotted with correspondence of the flow velocity. The current paper is mainly focused on

experimental studies of bacterial deposition pattern and pathogenesis. Thus, the mathematical models were intended to qualitatively assess the trend of experimental observations.

Reference

1. Cheryl A. Nickerson, C. Mark Ott, James W. Wilson, Rajee Ramamurthy, and Duane L. Pierson, Microbial Responses to Microgravity and Other Low-Shear Environments, *Microbiology and Molecular Biology Reviews*, Vol. 68, No. 2, 345-361(2004).
2. Joshua B. Gurtler, Nia A. Harlee, Amanda M. Smelser, Keith R. Schneider; *Salmonella enterica* Contamination of Market Fresh Tomatoes: A Review. *J Food Prot.* 2018; 81 (7): 1193–1213.
3. Oliveira, C., Carvalho, L. & Garcia, T. Experimental airborne transmission of *Salmonella Agona* and *Salmonella Typhimurium* in weaned pigs. *Epidemiology & Infection* 134, 199-209 (2006).
4. Wathes, C., Zaidan, W., Pearson, G., Hinton, M. & Todd, N. Aerosol infection of calves and mice with *Salmonella typhimurium*. *The Veterinary Record* 123, 590-594 (1988).
5. Harbaugh E, Trampel D, Wesley I, Hoff S, Griffith R, Hurd HS. Rapid aerosol transmission of *Salmonella* among turkeys in a simulated holding-shed environment. *Poult Sci.* 2006 Oct;85(10):1693-9.
6. Lahiri, A., Lahiri, A., Iyer, N., Das, P. & Chakravorty, D. Visiting the cell biology of *Salmonella* infection. *Microbes and infection* 12, 809-818 (2010).
7. Gantois, I., et al., Mechanisms of egg contamination by *Salmonella Enteritidis*. *FEMS microbiology reviews* 33, 718-738 (2009).
8. De Reu, K., et al., Eggshell factors influencing eggshell penetration and whole egg contamination by different bacteria, including *Salmonella enteritidis*. *International journal of food microbiology* 112, 253-260 (2006).
9. Kanamori H, Rutala WA, Weber DJ. The Role of Patient Care Items as a Fomite in Healthcare-Associated Outbreaks and Infection Prevention. *Clin Infect Dis.* 2017 Oct 15;65(8):1412-1419.
10. Moore, G., Blair, I. S. & McDOWELL, D. A. Recovery and transfer of *Salmonella typhimurium* from four different domestic food contact surfaces. *Journal of food protection* 70, 2273-2280 (2007).
11. Brutin, D., Starov, V.M., “Recent advances in droplet wetting and evaporation” *Chemical Society Reviews*, 2018.
12. Wang, Y., Liu, F., Yang, Y., Xu, L., “Droplet evaporation-induced analyte concentration toward sensitive biosensing”. *Mater. Chem. Front.*, 5, 5639-5652 (2021).
13. K. Sefiane Patterns from drying drops, *Adv. Colloid Interface Sci.*, 2014, 206 , 372 — 381.
14. R. D. Deegan Pattern formation in drying drops, *Phys. Rev. E: Stat. Phys., Plasmas, Fluids, Relat. Interdiscip. Top.*, 2000, **61** , 475 —485.
15. Robert D. Deegan , O. Bakajin , T. F. Dupont , G. Huber , S. R. Nagel and T. A. Witten , Contact line deposits in an evaporating drop, *Phys. Rev. E: Stat. Phys., Plasmas, Fluids, Relat. Interdiscip. Top.*, 2000, **62** , 756 —765.
16. R. D. Deegan , O. Bakajin , T. F. Dupont , G. Huber , S. R. Nagel and T. A. Witten , Capillary flow as the cause of ring stains from dried liquid drops, *Nature*, 1997, **389** , 829.

17. X. Zhong , A. Crivoi and F. Duan , Sessile nanofluid droplet drying, *Adv. Colloid Interface Sci.*, 2015, 217 , 13 —30.
18. T. S. Wong , T. H. Chen , X. Shen and C. M. Ho , Nanochromatography driven by the coffee ring effect, *Anal. Chem.*, 2011, 83 , 1871 —1873.
19. D. Kaya , V. A. Belyi and M. Muthukumar , Pattern formation in drying droplets of polyelectrolyte and salt, *J. Chem. Phys.*, 2010, 133 , 114905.
20. S. Tarafdar , Y. Y. Tarasevich , M. Dutta Choudhury , T. Dutta and D. Zang , Droplet Drying Patterns on Solid Substrates: From Hydrophilic to Superhydrophobic Contact to Levitating Drops, *Adv. Cond. Matter. Phys.*, 2018, 1 —24.
21. H. Hu and R. G. Larson , Marangoni Effect Reverses Coffee-Ring Depositions, *J. Phys. Chem. B*, 2006, 110 , 7090 —7094.
22. L. E. Scriven and C. V. Sternling , The marangoni effects, *Nature*, 1960, 187 , 186 —188.
23. R. G. Larson Re-shaping the coffee ring, *Angew. Chem., Int. Ed.*, 2012, 51 , 2546 —2548.
24. A. Rasheed, S. Sharma, P. Kabi, A. Saha, S. Chaudhuri, S. Basu, Precipitation dynamics of surrogate respiratory sessile droplets leading to possible fomites, *Journal of Colloids and Interface Science*, 600, 1-13 (2021).
25. Rocha, R., Almeida, C. & Azevedo, N. F. Influence of the fixation/permeabilization step on peptide nucleic acid fluorescence in situ hybridization (PNA-FISH) for the detection of bacteria. *PloS one* 13, e0196522 (2018).
26. Cefali, E., Patane, S., Arena, A., Saitta, G., Guglielmino, S., Cappello, S., Nicol`o, M. & Allegrini, M. Morphologic variations in bacteria under stress conditions: Near-field optical studies. *Scanning: The Journal of Scanning Microscopies* 24, 274-283 (2002).
27. Yunker, PJ, Still T, Lohr MA, and Yodh AG. "Suppression of the coffee-ring effect by shape-dependent capillary interactions." *Nature* 476, no. 7360 (2011): 308-311.
28. Kai Zhang, Liran Ma, Xuefeng Xu, Jianbin Luo, and Dan Guo, "Temperature distribution along the surface of evaporating droplets". *Phys. Rev. E* 89, 032404 (2014).
29. W. D. Ristenpart, P. G. Kim, C. Domingues, J. Wan, and H. A. Stone, "Influence of Substrate Conductivity on Circulation Reversal in Evaporating Drops". *Phys. Rev. Lett.* 99 (2007), 234502.
30. Thokchom, A.K., Medhi, B.J., Majumder, S.K. et al. Analysis of Circulation Reversal and Particle Transport in Evaporating Drops. *Microgravity Sci. Technol.* 33, 20 (2021).
31. Kjelleberg, S., Flardh, K., Nystrom, T. & Moriarty, D. Growth limitation and starvation of bacteria (Blackwell Scientific Publ., Oxford, UK, 1993).
32. Ottavio A. Croze, Gail P. Ferguson, Michael E. Cates, Wilson C.K. Poon, Migration of Chemotactic Bacteria in Soft Agar: Role of Gel Concentration, *Biophysical Journal*, 101, P525-534 (2011).
33. Ashish Kumar Thokchom, Rajaram Swaminathan, and Anugrah Singh, *Langmuir* 2014 30 (41), 12144-12153, DOI: 10.1021/la502491e.

REVIEWERS' COMMENTS:

Reviewer #1 (Remarks to the Author):

The authors have carefully considered the reviewer comments and modified the manuscript accordingly. After going through all the reply and the modified part of the manuscript, I believe that this manuscript is now suitable for publication.

Reviewer #2 (Remarks to the Author):

The authors have addressed the comments in the revised manuscript.